# The Chemical Profile, and Antidermatophytic, Anti-Inflammatory, Antioxidant and Antitumor Activities of *Withania chevalieri* A.E. Gonç. Ethanolic Extract

**DOI:** 10.3390/plants12132502

**Published:** 2023-06-30

**Authors:** Edmilson Emanuel Monteiro Correia, Artur Figueirinha, Lisa Rodrigues, José Pinela, Ricardo C. Calhelha, Lillian Barros, Chantal Fernandes, Lígia Salgueiro, Teresa Gonçalves

**Affiliations:** 1CNC-UC—Center for Neuroscience and Cell Biology of Coimbra, University of Coimbra, 3004-504 Coimbra, Portugal; edcorreia17@gmail.com (E.E.M.C.); lisa1cor@gmail.com (L.R.); xantal@gmail.com (C.F.); 2Faculty of Pharmacy, Health Sciences Campus, University of Coimbra, Azinhaga de S. Comba, 3000-548 Coimbra, Portugal; amfigueirinha@ff.uc.pt (A.F.); ligia@ff.uc.pt (L.S.); 3Associated Laboratory for Green Chemistry (LAQV) of the Network of Chemistry and Technology (REQUIMTE), University of Porto, 4099-002 Porto, Portugal; 4Centro de Investigação de Montanha (CIMO), Instituto Politécnico de Bragança, Campus de Santa Apolónia, 5300-253 Bragança, Portugal; jpinela@ipb.pt (J.P.); calhelha@ipb.pt (R.C.C.); lillian@ipb.pt (L.B.); 5Laboratório Associado para a Sustentabilidade e Tecnologia em Regiões de Montanha (SusTEC), Instituto Politécnico de Bragança, Campus de Santa Apolónia, 5300-253 Bragança, Portugal; 6Chemical Process Engineering and Forest Products Research Centre, Department of Chemical Engineering, Faculty of Sciences and Technology, University of Coimbra, 3030-790 Coimbra, Portugal; 7FMUC—Faculty of Medicine, University of Coimbra, Rua Larga, 3004-504 Coimbra, Portugal

**Keywords:** *Withania chevalieri*, Cape Verde, antidermatophytic activity, antioxidant activity, anti-inflammatory activity, anticancer activity, cytotoxicity

## Abstract

*Withania chevalieri*, endogenous from Cape Verde, is a medicinal plant used in ethnomedicine with a large spectrum of applications, such as treating skin fungal infections caused by dermatophytes. The aim of this work was to chemically characterize the *W. chevalieri* crude ethanolic extract (WcCEE), and evaluate its bioactivities as antidermatophytic, antioxidant, anti-inflammatory and anticancer, as well as its cytotoxicity. WcCEE was chemically characterized via HPLC–MS. The minimal inhibitory concentration, minimal fungicidal concentration, time-kill and checkerboard assays were used to study the antidermatophytic activity of WcCEE. As an approach to the mechanism of action, the cell wall components, β-1,3-glucan and chitin, and cell membrane ergosterol were quantified. Transmission electron microscopy (TEM) allowed for the study of the fungal ultrastructure. WcCEE contained phenolic acids, flavonoids and terpenes. It had a concentration-dependent fungicidal activity, not inducing relevant resistance, and was endowed with synergistic effects, especially terbinafine. TEM showed severely damaged fungi; the cell membrane and cell wall components levels had slight modifications. The extract had antioxidant, anti-inflammatory and anti-cancer activities, with low toxicity to non-tumoral cell lines. The results demonstrated the potential of WcCEE as an antidermatophytic agent, with antioxidant, anti-inflammatory and anticancer activity, to be safely used in pharmaceutical and dermocosmetic applications.

## 1. Introduction

Natural medicine has always played an important role in human health care. The World Health Organization Global Centre for Traditional Medicine (https://www.who.int/initiatives/who-global-centre-for-traditional-medicine, accessed on 20 January 2023) estimates that about 88% of all countries use traditional medicine involving the use of natural products, especially medicinal plants and their various extracts easier to obtain and use, as a source of compounds with potential for disease prevention and treatment [1]. Therefore, an increasing number of researchers are involved in the intensive screening of plants used in traditional medicine to discover new bioactive molecules with a therapeutic potential [2,3]. There are numerous plants scientifically documented as having in vitro, ex vivo and in vivo effectiveness in the treatment of cutaneous diseases, as alternatives to conventional treatments [2,4,5].

Plants of the genus *Withania* (Solanaceae), such as *Withania somnifera* (L.) Dunal (Ashwagandha) and *Withania coagulans* (Stocks) Dunal [6], are widely distributed across drier regions of tropical and subtropical zones, from the Canary Islands through to the Mediterranean region, and from North Africa to Southwest Asia. In these regions, these plants are widely used in folk medicine and claimed to have biological properties attributed to their components such as triterpenoids, steroid lactones known as withanolides, and also phenolic compounds. Data have been gathered indicating that these compounds exert protective effects on skin diseases, oxidative stress, inflammatory disorders, cancer and microbial diseases, among others [7,8,9,10,11,12,13,14]. Although the medicinal use of some *Whitania* species, particularly *W. somnifera*, have been recognized for centuries, there is a constant update on the recognition of other species of this genus with interesting bioactivities. The phytochemical composition of the root extract from *Withania aristata* (Aiton) Pauquy, endemic in the North African Sahara, has recently been described, along with its strong antimicrobial activity against resistant bacteria and fungi, as well as its antioxidant and anti-inflammatory effects [15].

In Cape Verde, the endemic *Withania chevalieri* A.E. Gonç. is commonly known as a chili pepper, growing only on the islands of São Vicente, Sal and Fogo [16,17]. From this plant, crude ethanolic extracts are prepared and used in Cape Verdean traditional medicine for its broad-spectrum biological properties [17,18]. Based on ethnopharmacological information, homemade formulations of *W. chevalieri*, including tinctures (dried plant soaked in 70% ethanol) or ointments (powder of plant mixed with sterile petroleum jelly), are widely used as antimicrobials, particularly for fungal skin infections such as dermatophytoses, and in skin ulcers due to infections by several bacteria. These formulations are also used as anti-inflammatory and anticancer agents. However, there are no scientific data on the biological activities and chemical composition of these extracts. Therefore, the objectives of this work were to chemically characterize the *W. chevalieri* ethanolic extract (WcCEE) and evaluate its biological properties, supporting traditional uses of this plant.

## 2. Results

### 2.1. Phytochemical Profile of the Crude Ethanol Extract

The WcCEE chemical analysis was performed using high-performance liquid chromatography coupled with a photodiode array detector, and a mass spectrometry detector with an electrospray interface (HPLC–PDA–ESI–MS^n^) (Figure 1).

All chromatographic and spectral data, essential for the identification of compounds, are included in Table 1. The identification was performed using the absorption maxima values of the ultra violet/visible (UV/vis) spectra (λmax) that are normally characteristic of each phytochemical class, complemented with information obtained by mass spectrometry (ESI–MSn), namely the mass of the precursor ion, that allows for inferring the molecular mass of the compounds and its successive fragmentations (MS2 and MS3) related to the molecular structure. The analysis of these data, made for each compound, is described below and revealed the presence of several classes of compounds, namely organic acids (citric acid), aminoacids (phenylalanine), phenolic compounds such as hydroxycinnamic acid derivatives and flavonoid glycosides, and a group of terpenes called withanolides (Table 1).

#### 2.1.1. Hydroxycinnamic Acids

Peaks 3 and 4 exhibited UV spectra with a maxima at 324 nm and a shoulder near 292 nm, suggesting the presence of caffeic or ferulic acid derivatives. The MS profile of peak 3 was characterized by a pseudomolecular ion at *m*/*z* 353, with a base peak at *m*/*z* 191 in the MS^2^ spectrum, which is characteristic of 5-O-caffeoylquinic acid [16]. Peak 4 was tentatively identified as 4-O-feruloylquinic acid because of its mass spectrum containing a molecular ion at *m*/*z* 367 and an MS^2^ base peak at *m*/*z* 173, which is similar to the pattern described by other authors for the same structure [19].

#### 2.1.2. Hydroxycinnamic Amides

Peaks 5, 8, 10 and 11 were identified as hydroxycinnamic amides based in their UV and mass spectral behavior. Peak 5 exhibited a UV maxima at 318 nm and 294 nm, and displayed a pseudomolecular ion [M+H]^+^ at *m*/*z* of 251 and characteristic fragment ions at *m*/*z* 234, 163 and 89, suggesting the presence of caffeoylputrescine [20,21]. Peak 8 was identified as feruloyltyramine, since it presented UV spectra consistent with the presence of a feruloyl moiety and a characteristic fragmentation in the mass spectra, as described by Bolleddula et al. [22] and Nikolić et al. [23]. Peak 11 differed from peak 8 by the loss of a methyl group. Thus, peak 8 was identified as methoxyferuloyltyramine. Peak 10 was identified as bis(dihydrocaffeoyl)spermidine, according to the fragmentation pattern proposed by Li et al. [24].

#### 2.1.3. O-Glycosylflavonoids

Peaks 6, 7 and 9 exhibited UV spectra characteristic of 3-O-subtituited flavonols, namely two absorption bands: band II between 257 and 272 nm, and band I near 340–350 nm, the former being the most intense band. The three compounds presented the same molecular ion at *m*/*z* 609 and showed the loss of a rutinoside moiety [(M–H)- 162–146]^−^, originating fragment at *m*/*z* 301, which in turn originated fragments 179 and 151, characteristic of a quercetin aglycon. These results suggest that peaks 7 and 9 are probably quercetin-*O*-hexosyl-deoxyhexoside isomers, while the compound of peak 6 had the same structure with an additional hexosyl unit [25]. The UV spectrum of peak 12 is characteristic of flavones, exhibiting absorption band I at 350 nm and band II in the 254–272 nm range. In first-order MS, the pseudomolecular ion at *m*/*z* 593 was followed by the loss of a rutinoside moiety [(M–H)- 162–146]^−^, originating the intense signal at *m*/*z* 285 due to luteolin aglycon. Therefore, luteolin-*O*-hexosyl-deoxyhexoside was the proposed structure for the compound in peak 12 [26].

#### 2.1.4. Withanolides

Several withanolides were identified in the extract due to their UV absorption maxima near 230 nm, which is consistent with the presence of an α, β-unsaturated δ-lactone [27]. Peak 13 exhibited a pseudomolecular ion at *m*/*z* 547 [M–H]^−^ and originated fragments at *m*/*z* 501 [M–H–HCOOH]^–^, *m*/*z* 483 [M–H^+^–HCOOH–H_2_O]^–^ and *m*/*z* 179 [M–H^+^–HCOOH–H^2^O–C_17_H_20_O_5_]^–^. This fragmentation pattern is similar to baimantuoluoside J [28]. Compounds 14 and 15 were identified as withanolide diglucoside isomers, probably withanoside II isomers, because they exhibited a pseudomolecular [M+H]^+^ ion at **m*/*z** 799 and originated fragments from the loss of glucose units at *m*/*z* 637 and 475 [22]. Peak 17 was identified as withanolide S, since its mass fragmentation pattern in a positive ion mode consisted of a signal adduct [M+NH_4_]^+^ at *m*/*z* 522, characteristic losses of water molecules and fragments in second-order spectra at *m*/*z* 505, 488, 319 and 169. The diagnostic signal at *m*/*z* 169 resulted from the cleavage of C17 and C20 as a result of the presence of hydroxylation at C20 [29]. The fragmentation patterns exhibited by peaks 18 and 19 were similar to patterns exhibited by ashwagandhanolide (MW 974) and withanolide sulfoxide (MW 992), respectively, two withanolide dimers previously found in the roots of *Withania somnifera* [29]. Peaks 20 and 21 exhibited a pseudomolecular ion at *m*/*z* 783 in positive ion mode mass spectra, together with an ammonium adduct [M+NH4]^+^ at *m*/*z* 800, which is common in diglycosilated withanosides IV, VI and X, and also withanamides D and E. However, in the case of peak 20, the loss of a lactone moiety (−172 amu) from the fragment at *m*/*z* 441 originates the signal at *m*/*z* 269, which is consistent with the presence of withanoside IV, while in peak 21, the presence of the signal at *m*/*z* 169 indicates hydroxylation at C20, suggesting the presence of withanoside VI [22,29,30]. Compounds 22, 24, 25 and 27 exhibited similar fragmentation patterns in a positive ion mode, despite some minor differences in signal abundances. The characteristic adducts at *m*/*z* 488 [M+NH_4_]^+^, 493 [M+Na]^+^, 941 [2M+H]^+^ and 958 [2M+NH_4_]^+^, and characteristic water losses at *m*/*z* 453 [M+H_2_O]^+^, 435 [M+2H_2_O]^+^, 417 [M+3H_2_O]^+^ and 399 [M+4H_2_O]^+^ were observed in mass spectra. This behavior was previously described for several structures of withanolides, withaferins, withanones and withacoagulins [29]. In a negative ion mode, these compounds showed a deprotonated molecular ion at *m*/*z* 469 [M–H]^−^ or a related adduct, and characteristic fragments generated from water loss. The compounds 22, 24 and 27 exhibited the loss of a δ-lactol unit (C_7_H_10_O_3_) ( -142 a.m.u), while peak 25 showed the loss of a δ-lactone unit (C_7_H_10_O_2_) (- 126 a.m.u.). Based on these results, compounds 22, 24 and 27 could be identified as a type-A δ-lactol withanolide and compound 25 a type-A δ-lactone withanolide. Peak 23 displayed a pseudomolecular ion at *m*/*z* 621 [M+H]^+^, of which the fragmentation generated fragments at *m*/*z* 459, 441 and 423, which are characteristic of coagulin Q/physagulin D [26]. Peak 28 exhibited an apseudomolecular ion at *m*/*z* 475 in a positive ion mode. The fragmentation originated signals at *m*/*z* 457, 439 and 421. Based on the fragmentation pattern proposed by Bolleddula et al. [22], peak 28 was identified as withanoside II aglycone. Both compounds 29 and 30 showed the same spectral behavior, with the UV spectra maximum at 228 nm and an intense signal at *m*/*z* 499 [M+HCOO-] in a negative ion mode. In a positive mode, both compounds exhibited fragments at *m*/*z* 472 [M+NH_4_]^+^, 477 [M+Na]^+^ and 926 [2M+NH_4_]^+^. In MS^2^, the fragments observed were at *m*/*z* 453, 437, 267 and 169, which is consistent with the pattern previously described for withacoagulins [29]. Therefore, the identification proposed for compounds 29 and 30 was withacoagulin isomers.

#### 2.1.5. Other Compounds

Peak 1 was identified as citric acid due to its pseudomolecular ion [M−H]^−^ at *m*/*z* 191 and fragmentation pattern that originated a base peak at MS^2^ at *m*/*z* 111, and also a characteristic fragment at *m*/*z* 173 [31]. Peak 2 showed a pseudomolecular ion at *m*/*z* 166 and a major fragment in MS^2^ at *m*/*z* 120. This spectral behavior, along with the observed UV maximum at 269 nm is consistent with the presence of phenylalanine [23]. Peak 16 showed a molecular ion at *m*/*z* 625 and fragments in MSn at *m*/*z* 488 and 351, consistent with consecutive losses of tyramine units (−177 a.m.u.). This behavior was previously reported for grossamides [22].

### 2.2. Antifungal Activity

The antifungal activity of WcCEE was performed using the EUCAST protocol. Since we used the clinical isolates of several dermatophyte species, we also quantified the susceptibility of the isolates to three conventional antifungals clinically used in dermatophytosis, itraconazole, terbinafine and griseofulvin. The results (Table 2) obtained show that some isolates of *M. canis* (PT01), *T. soudanense* (CV3, CV20, CV24, CV45, CV47, CV54) and *T. rubrum* (CV55) were resistant to griseofulvin because the minimal inhibitory concentrations (MIC) values for griseofulvin were higher than 3 µg/mL [32].

The determination of the MIC and the MFC revealed moderate-to-strong fungicidal activity of WcCEE on all the tested clinical isolates of dermatophytes, with MFC = MIC or MFC = 2 × MIC. The lowest WcCEE MIC value was determined for some *T. soudanense* isolates (1.56 mg/mL), while for *T. rubrum* and *T. interdigitale*, the lowest MIC value determined was 3.12 mg/mL (Table 2). *M. canis* isolates showed a low susceptibility to WcCEE, requiring a concentration of 25 mg/mL of WcCEE for considerable growth inhibition. The control, with residual ethanol concentrations, did not inhibit the growth of dermatophytes (data not shown).

Regarding the fungicidal kinetics of WcCEE (Figure 2), it was observed that for MIC and 2 × MIC values, the WcCEE showed a concentration-dependent fungicidal effect (>99.9% CFU reduction) after 72 h of incubation in all dermatophyte strains except *T. interdigitale* (Figure 2c), in which 2 × MIC had fungicidal action at 48 h, and for *M. canis* (Figure 2d), at 96 h. For concentrations of 4 × MIC, after 24 h of exposure, WcCEE exerted a fungicidal effect on *T. interdigitale* (Figure 2c) and *M. canis* (Figure 2d), and *T. soudanense* (Figure 2a) and *T. rubrum* (Figure 2b) after 48 h. Despite significantly reducing fungal growth (*p* < 0.05), after 48 h of exposure, subinhibitory concentrations (MIC/2) did not completely inhibit growth at the end of 5 days of exposure, when compared to the growth control (no WcCEE added).

Checkerboard assays allowed us to study the possible synergies between WcCEE and antifungals commonly used in the treatment of dermatophytoses. A selected number of dermatophyte isolates were used, and the results showed a synergistic effect in all the isolates tested when the WcCEE was combined with terbinafine; when WcCEE was combined with itraconazole, synergy was observed in 4/10 isolates, while the combination of WcCEE with griseofulvin led to synergy in 2/10 isolates (Table 3). No antagonistic effects were observed.

To exclude the possibility of the induction of resistance by WcCEE, dermatophyte isolates were repeatedly exposed to the extract. The MIC values for *T. soudanense* and *T. rubrum* remained unchanged after 15 serial passages, whereas for *T. interdigitale*, the MIC value doubled after 13 passages (Table 4). Nevertheless, the increase in MIC was not higher than 3 times for any of the isolates.

### 2.3. Effect of WcCEE on the Ultrastructure of Dermatophytes

Ultrastructural changes were studied on selected strains of *T. rubrum*, *T. interdigitale* and *T. soudanense* in response to WcCEE, using TEM. Growth in the presence of WcCEE led to ultrastructural modifications when compared with the control. The cell wall thickness (Figure 3g) decreased (*p* > 0.05; for *T. soudanense*) and, when compared with the controls (Figure 3a,c,e), the cell wall became less organized, looking more electrodense when the fungi were exposed to WcCEE (Figure 3b,d,f). Other modifications were observed in the cytoplasm, with big vacuoles concentrating the cytosol in the cell periphery against the cell membrane and the cell wall (Figure 3i,j). In some of the vacuoles, it was possible to observe intra-vacuole double membrane systems, some of which can be described as autophagic-like swirls (Figure 3k,l). These ultrastructural observations were clearly different from the controls, with regular cell walls with organized cellular compartments, particularly mitochondria (Figure 3a,c,e). Some fungal cells appeared as “ghosts”, intact cell walls with a hardly organized content and with cell membranes severely destroyed (Figure 3b,d,f). This led us to include in the TEM study a control for *T. rubrum* in the presence of itraconazole, an antifungal of the class of azoles interfering with azole synthesis and leading to membrane damage (Figure 3h). This confirmed that all three species had ultrastructural modifications with signs of severe damage in the cell membrane when exposed to WcCEE, similar to what was observed with itraconazole-exposed fungi.

In what regards the contents of cell membrane ergosterol, the results showed that for all the fungi, there was an increase due to WcCEE exposure when compared with control conditions, but only with statistical significance (*p* < 0.001) for *T. soudanense* and *T. interdigitale* (Figure 4a). Exposure to WcCEE led to a slight increase in the cell wall components, β-1,3-glucan (Figure 4b) and chitin (Figure 4c), not statistically significant (*p* > 0.05), except for *T. soudanense* (Figure 4c), in which a slight decrease in chitin was observed.

### 2.4. Antioxidant Activity

The 50% inhibitory concentration (IC_50_), defined as the concentration required to obtain half of the 100% maximum antioxidant effect, was evaluated based on the results of the TBARS and OxHLIA methods. The IC_50_ values in these cell-based assays were 2.1 mg/mL and 0.49 mg/mL for TBARS and OxHLIA methods, respectively, as indicated in Table 5. The antioxidant activity of the extract for the OxHLIA and TBARS was lower than that quantified for the standard control, Trolox (0.0091 mg/mL for the TBARS and 0.0218 mg/mL for the OxHLIA methods).

The antioxidant activity of the extract (cellular antioxidant activity—CAA) was quantified in RAW 264.7 cells by examining the percentage of reduction in fluorescence. Fluorescence was reduced by up to 60% when in the presence of the extract, and up to 95.3% when in the presence of quercetin, as shown in Table 5.

### 2.5. Anti-Inflammatory Activity

The extract showed an inhibitory effect on the NO production in LPS-activated RAW 264.7, with an IC_50_ of 7 µg/mL, meaning that the NO production was at a low concentration, similar to that obtained in dexamethasone, 6.3 µg/mL (Table 5).

### 2.6. Cytotoxicity in Tumor and Non-Tumor Cells

To determine whether the extract can be associated with anticancer activity, the cytotoxic effect in human tumor cell lines, AGS (gastric adenocarcinoma), CaCo2 (colorectal adenocarcinoma), MCF-7 (breast adenocarcinoma) and NCI-H460 (lung carcinoma) was assessed. It was found that the extract was cytotoxic to all cancer cells tested at low concentrations. Lung carcinoma (NCI-H460) and breast adenocarcinoma (MCF-7) were more susceptible to the extract, with GI_50_ (concentration of the extract that inhibits cell growth by 50%) values of 19 and 27 µg/mL, respectively, followed by gastric adenocarcinoma human (AGS) and colorectal adenocarcinoma (CaCo2), with GI_50_ values of 47 and 63 µg/mL, respectively. These results are indicative of the potential of WcCEE as an inhibitor of the growth of human cancer cells, especially with NCI-H460 cells. The cytotoxicity assays of the extract in non-tumor cell lines Vero (African green monkey kidney) and PLP2 (primary porcine liver cell culture) showed a lower effect, with a GI_50_ value > 400 µg/mL, compared to the control, ellipticine (Table 5). Additionally, the results showed that WcCEE did not affect the viability of the HaCat keratinocyte cell line, with the lowest concentration tested up to 1.56 mg/mL (Appendix A).

## 3. Discussion

In this work, we characterized for the first time the composition of an ethanolic extract of *W. chevalieri*, an endemic plant of Cape Verde used in traditional medicine, and evaluated several bioactivities, such as antidermatophytic, antioxidant, anti-inflamatory and anticancer activities, using clinical isolates. To date, few *Withania* species have been chemically characterized, with *W. somnifera* and *W. coagulans* being the most studied by far. Recent studies of different extracts analyzed using different methods revealed withanolides as being the major constituents of *W. somnifera,* along with different classes of withanosides (4-OH and 5,6-epoxy withanolides (withaferin A-like steroids) and 5-OH and 6,7-epoxy withanolides (withanolides A-like steroids) [33]. Lipids, sugars, amino acids, organic acids, phenolic compounds, flavonoids and many other secondary metabolites with a broad spectrum of activity were also found in these plants [13]. WcCEE was characterized by organic acids (in particular citric acid), essential amino acids (phenylalanine), phenolic acids (5-*O*-caffeoylquinic acid and 4-*O*-feruloylquinic acid)) flavonoids (quercetin and luteolin), phenolamides (bis(dihydro caffeoyl)spermidine, caffeoyl putrescine, feruloyl tyramine, and methoxyferuloyltyramine)), withanolides and other active compounds, all known to be important compounds with beneficial medicinal properties [10,34,35]. In general, the compounds present are similar to those found in *Withania somnifera* and other species of the genus *Withania*; however, our study revealed the presence of ashagandhanolide and withanolid sulfoxide, previously found only in the roots of *Withania somnifera* [29] and baiantuoluoside J, a norwithasteroid only found in the *Datura* species [28]. Therefore, we consider that the results of the phytochemical characterization of WcCEE represent an important contribution to the expansion of knowledge about the species of the genus *Withania*.

All the clinical isolates tested were susceptible to the antifungal activity of WcCEE, with a MIC range of 1.56–25 mg/mL, even the strains resistant to griseofulvin. We used several clinical isolates of different dermatophyte species, some of which were isolated in Cape Verde. There are no previous studies on the antimicrobial activity of *W. chevalieri* and the studies on the antidermatophytic activity of extracts obtained from other *Whitania* species are scarce, although the activity against other fungi was previously demonstrated (e.g., [15,36,37,38,39,40]). When compared with other *Withania* species, *W. somnifera* and *W. coagulans*, the MIC values obtained had similar ranges. In fact, extracts of *W. somnifera* were shown to have an antifungal effect against other species of dermatophytes, *T. mentagrophytes* and *M. gypseum*, with MIC values ranging from 1.56 to 3.12 mg/mL [37], and MIC values up to twelve times higher for *T. mentagrophytes* and *E. floccosum* [38]. However, another study showed that for a *W. somnifera* seed extract, the MIC values against *T. rubrum* and *Trichophyton tonsurans* were lower [41]. To further complement this, we also assessed whether the antifungal effect was fungicidal or fungistatic and determined the minimal fungicidal concentration, i.e., MFC [42]. This fungicidal effect was also confirmed by time-kill assays, which showed that WcCEE exerted a dose-dependent fungicidal activity after 48 to 72 h of exposition. Moreover, using the checkerboard assay, we were able to prove the synergistic effect of WcCEE when combined with terbinafine. Since the combinations of drugs with different mechanisms of action that result in synergy represent an efficient therapeutic alternative [43,44], it can be concluded that WcCEE has the potential to be used in combination with other antifungals to increase the efficacy of conventional antidermatophytic therapies. Resistance is a major concern, and to address this problem, a more responsible use of current drugs and the development or discovery of new antifungal agents are needed [45]. Our results also clearly demonstrate that WcCEE does not induces resistance after repeated exposure. In fact, this is an important issue when seeking novel antimicrobial therapies, and we believe this is the first study demonstrating that an extract from plants of the genus *Whitania* do not induce resistance in fungi, specifically in dermatophytes. This might result from the chemical complexity of the extract acting on diverse targets, as described by others [46].

To identify the mechanism of action underlying the antifungal effect of the extract, the cell membrane and cell wall components were quantified because these are important targets for currently approved antifungals [47], whereby it was observed that β-1,3-glucan, chitin and ergosterol were increased. The increase in ergosterol does not corroborate the report by Bitencourt et al. [48], outlining that in *T. rubrum*, quercetin, one of the compounds also present in WcCEE, downregulates the fatty acid synthase reducing ergosterol levels, thereby causing plasma membrane disruption. Several studies have demonstrated that the increased expression of genes coding for enzymes responsible for the synthesis of cell wall components is a compensatory mechanism in response to exposure to antifungals (paradoxical effect) [49,50]. In this study, WcCEE induced a slight increase in the cell wall component. This result, together with the fact that certain compounds found in WcCEE can induce cell lysis [51], led us to speculate that one of the possible mechanisms of action of WcCEE as an antidermatophytic might include plasma membrane lysis. Moreover, the ultrastructural modifications observed indicate the effectiveness of WcCEE in causing damage to fungal cells. One of the most notable abnormalities is the presence of large vacuoles confining the cytoplasmic content against the cell membrane and cell wall, a sign of stressful conditions [52]. Several studies have reported the ability of synthetic compounds [53] or natural compounds, including withaferin A isolated from *W. somnifera* [54], to induce cytoplasmic vacuolization.

In this work, three methods were used to assess the antioxidative activity contemplating the two criteria proposed by Laguerre et al. [55], and according to Phongpaichit et al. [56], WcCEE showed a strong antioxidant activity. This result is in line with what was reported for ethanolic extracts of the root and leaves of *Withania frutescens* [39], ethanolic extracts of *W. somnifera* [7], hydro-ethanolic extract of *W. aristata* [15] and ethyl-acetate extracts of *W. somnifera* seeds [41], with a higher antioxidant activity than those of aqueous or hydroethanolic extracts [7].

Inflammation is a response to tissue damage which, when exacerbated, remains a devastating and challenging health problem [57]. In this study, the anti-inflammatory activity of WcCEE was demonstrated to be similar to dexamethasone, indicating that this extract can be used to decrease or prevent exacerbated inflammation and cellular damage. This beneficial bioactivity was described for other *Withania* spp., both in vitro and in vivo [34,41,58].

The extract showed a remarkable in vitro inhibitory growth of several human cancer cell lines, especially lung cancer (NCI-H460). Earlier, Jayaprakasam et al. [35] reported a similar trend in growth inhibition, with extracts of *W. somnifera* leaves exhibiting higher cytotoxic properties against lung (NCI-H460) than breast and colon cancer cell lines.

All these results strongly indicate that *W. chevalieri* ethanolic extracts work as an efficient antioxidant, anti-inflammatory and cytotoxic agent, and it can be speculated that these activities are due to synergies between polyphenolic compounds present in WcCEE. In fact, polyphenol-rich medicinal plants are known for their biological properties. The total content of polyphenols in WcCEE (mainly flavonoids, phenolic acids and phenolamides) corroborates other studies focused on the potential of phenolic-rich extracts having in vitro antioxidant, anti-inflammatory and cytotoxic activities [59,60]. The results obtained demonstrate the absence of toxicity in different non-tumor cell lines, including human skin cells and the HaCat cell line (Appendix A), which, together with an antifungal effect against dermatophytes, opens the possibility of the safe use of *W. chevalieri* extracts as a topical treatment for superficial infectious diseases.

## 4. Materials and Methods

### 4.1. Extract Preparation

The plants (Figure 5a) were collected in the volcanic area of Fogo Island, at Chã das Caldeiras (latitude: 14°56′00″; longitude: −24°22′00″; altitude: 700 m), between 29 August and 6 September 2021. These were taxonomically identified as *W. chevalieri* by Dr. Isildo Gomes, a botanist working at the National Institute of Agricultural Research and Development of Cape Verde. After being washed with water to remove dust, the plants were dried (Figure 5b) outdoors for 14 days, and 5 more days in an incubator at 50 °C to remove residual moisture. The whole plant (roots, stem, fruits, flower and leaves) was ground into powder. The extraction procedure was the closest to the traditional process. To initiate the ethanolic extraction, 10 g of the powder was mixed with 100 mL of 70% ethanol, and incubated at 25 °C for 4 days in a shaker, at 150 rpm. This suspension was filtered using gauze and a Buchner funnel, followed by filtration using a common funnel with gauze, and evaporated at 60 °C for 24 h, always under aseptic conditions. The extraction yield was 5.4%. The stock of WcCEE obtained was stored at 4 °C for more than 6 months, without losing its activity (data not shown). The dry extract was resuspended in 20% ethanol, filtered through Whatman filters (0.4 and then 0.22 µm), and used for the screening of bioactivities and chemical characterization.

### 4.2. Dermatophyte Isolates and Fungal Culture Conditions

A total of 26 different clinical isolates from the CYC-UC fungal collection (University of Coimbra) were used (Table 6). These fungi were grown at 30 °C on a potato dextrose agar (PDA, Difco) for 7 days, except *M. canis*, which was cultured on a rice agar medium for 14 days.

### 4.3. Chemical Characterization of the Crude Ethanolic Extract

The chemical characterization of WcCEE was performed using a liquid chromatograph with a photodiode spectrophotometer PDA detector (Thermo Finnigan Surveyor, San Diego, CA, USA), interfaced with a linear ion trap mass spectrometer (LIT–MS) (LTQ XL, Thermo Scientific, Waltham, MA, USA). The LC column was a Waters Spherisorb ODS2 (3 µm, 150 × 2.1 mm) (Waters Corporation, Milford, MA, USA), and separation was carried out at 20 °C. Of the dissolved extract (2.65 mg/mL in methanol 50%), 20 µL was injected and the elution was performed with a mobile phase consisting of 2% aqueous formic acid (solvent A) and methanol (solvent B). Formic acid and methanol (HPLC grade) were purchased from Merck (Lisbon, Portugal). The gradient profile used was 0–70 min, 20–100% B. The flow rate was 200 µL/min. The first detection was made by the diode array spectrophotometer between 220 and 650 nm.

The second detection was made by the mass spectrometer in positive and negative electrospray ionization (ESI) modes, and was programmed to perform a series of three scans: a full mass (MS), and a MS^2^ and MS^3^ of the most abundant ion. The collision gas was helium, with a normalized collision energy of 35%. Nitrogen was used as the nebulizer gas, with a sheath gas flow of 35 (arbitrary unit) and an auxiliary gas flow of 20 (arbitrary unit) in the negative mode, and with a sheath gas flow of 40 (arbitrary unit) and an auxiliary gas flow of 5 (arbitrary unit) in the positive mode. The capillary temperature and source voltage were set at 275 °C and 5.00 kV, respectively. The capillary voltage was set at −35.00 V for the negative mode, and 40.00 V for the positive mode.

### 4.4. Antifungal Susceptibility Assays

#### 4.4.1. Preparation of Inocula and Antifungal Solutions

Inocula were prepared according to the European Committee on Antimicrobial Susceptibility Testing (EUCAST) definitive document, E.DEF 11.0 [61], with slight modifications. The cultures were covered with 2 mL of sterile saline (0.85%) and scraped using the tip of a sterile swab; this mixture was collected into sterile tubes. Afterward, the tubes were vortexed to detach the conidia from the hyphae and allowed to rest for 5 min for the sedimentation of the hyphae. Homogeneous conidial suspensions were transferred to new sterile tubes, and the number of cells was counted using a hemocytometer chamber and adjusted to 2 × 10^6^–5 × 10^6^ UFC/mL. The final working inoculum was first prepared by diluting the preceding inoculum to 1:10 in 0.85% saline solution, and then adjusted to a concentration of 2 × 10^5^ UFC/mL in 2 × RPMI 1640, supplemented with chloramphenicol (50 mg/mL).

The antifungal solutions, itraconazole (Actavis, 100 mg capsules; purity, 25%), terbinafine (Acros Organics, Fisher Scientific, Hampton, NH, USA), and griseofulvin (Sigma Aldrich, St. Louis, MI, USA,) were prepared at the concentration stock recommended by EUCAST standards.

#### 4.4.2. MIC and MFC Determination

The minimum inhibitory concentrations (MICs) of WcCEE and conventional antidermatophytic agents were determined using the broth microdilution method, according to the EUCAST definitive document, E.DEF 11.0 [61]. Twofold serial dilutions were performed on the plate to obtain concentrations ranging from 0.097 to 50 mg/mL for WcCEE, 0.001 to 1 µg/mL for itraconazole and terbinafine, and 0.031 to 16 µg/mL for griseofulvin. After adding 100 µL of the inoculum in a two-fold concentration of RPMI, the microplates were incubated at 35 °C and visually read after 96 h of incubation. The MIC was defined as the lowest concentration at which there was no visible growth after 4 days of incubation at 35 °C. To evaluate if the WcCEE extract was fungicidal or fungistatic against dermatophytes, the minimum fungicidal concentration (MFC) was determined. After the MIC determination, 30 µL of the content of the wells corresponding to the MIC, the 2 × MIC, 4 × MIC and to 8 × MIC were cultured on PDA plates and rice agar for 7 days, at 30 °C. The MFC is the minimum concentration of the antifungal drug resulting in a 99.9% reduction in fungal cell counts compared to the starting inoculum [62]. Then, the ratio of MFC–MIC was determined. An antifungal agent is considered fungicidal if this ratio does not exceed a value of 4, otherwise it is considered fungistatic [63,64]. These assays included a control, with residual 5% ethanol.

#### 4.4.3. Time-Kill Assay

The time-kill kinetics of WcCEE on the tested strains of dermatophytes was carried out following the procedure described by Lana et al. [65], with some modifications. Concentrations of 0.5 × MIC, MIC, 2 × MIC and 4 × MIC of the extracts were prepared. For this assay, one strain of *T. soudanense*, *T. rubrum* and *T. interdigitale* was selected. Inocula of 1 × 10^5^ CFU/mL were diluted in RPMI to a concentration of 1 × 10^4^ CFU/mL, for a final volume of 7 mL, containing WcCEE at desired concentrations. After 4 days of incubation at 35 °C, aliquots of 0.1 mL were taken from the tubes at different time intervals (0, 24, 48, 72 and 96 h), and diluted to obtain 10–10^3^ dilutions. From these, aliquots of 30 μL were taken and spread on PDA plates, and incubated at 30 °C for 7 days. A growth control was included for each organism without the extracts.

#### 4.4.4. Checkerboard Assay

The impact of WcCEE combination with clinically used antidermatophytic agents (itraconazole, terbinafine and griseofulvin) was evaluated using the checkerboard assay method [66]. Selected isolates of *T. soudanense* (3), *T. rubrum* (3), *T. interdigitale* (2) and *M. canis* (2) were used. To quantify the interactions between WcCEE and the tested antifungal agents, the fractional inhibitory concentration index (FICI) value was calculated, as described by others [66]. The impact on the drug potency was defined as synergistic when FICI ≤ 0.5, additive when 0.5 < FICI ≤ 4 and antagonist when FICI ≥ 4 [67].

#### 4.4.5. Development of Resistance

The assay to evaluate the possibility of dermatophytes to develop resistance against WcCEE was carried out using the serial passages method described by Ghannoum et al. [68]. From the isolates used in the MIC–MFC assay, one strain of *T. soudanense, T. mentagrophytes* and *T. rubrum* was selected to determine whether repeated exposure to crude WcCEE would cause the development of resistance. After determination of the MIC, the contents of sub-MIC wells were subcultured on PDA and rice agar plates for 7 days. Afterward, 7 mL of the inocula (1 × 10^5^ CFU/mL) prepared from these subcultures was incubated for 5 days at 35 °C, with a WcCEE equivalent to 0.5 × MIC. The tubes were then centrifuged at 16,000× *g* for 3 min, and the sediment was subcultured. After 7 days of incubation, this subculture was subsequently used to repeat the MIC assays under the conditions described above, to determine whether an increase in MIC occurred. This whole process was repeated 15 times (independent experiments) in duplicate. A MIC increase higher than 3 times the initial MIC was considered an indication of resistance development.

### 4.5. Effect of WcCEE in Ergosterol and Cell Wall Components, and in Fungal Ultrastructure

Changes associated with the cell wall components and cell membrane due to WcCEE exposure were studied through the quantification of chitin and β-(1,3)-glucan and ergosterol, respectively. Ultrastructural modifications were studied using transmission electron microscopy (TEM) analysis. The fungal growth and preparation of mycelia were performed according to Fernandes et al. [50], with minor modifications. Mycelial growth was prepared by inoculating 0.5 mL of cell suspensions (2 × 10^7^ CFU/mL) in 100 mL of a yeast malt extract (YME) liquid medium (growth control), and YME containing WcCEE and itraconazole as the positive control. The cultures were incubated with shaking at 120 rpm, at 30 °C for 5 days.

For TEM analysis, mycelium balls, formed after 5 days of incubation, were recovered and fixed with 2.5% glutaraldehyde in 0.1 M sodium cacodylate (pH 7.2). Post-fixation was performed as reported previously [66]. The samples were observed using a transmission electron microscope (FEI-Tecnai^®^ G2 Spirit Bio TwinTM, Tartu linn, Tartumaa) at 100 kV. For cell wall thickness quantification of each tested fungus, eight TEM images were selected: four different cell sections for the controls and four for the samples exposed to the extract. Of these, ten measurements of each were collected, and the averages and standard deviations were calculated.

For the quantification of chitin, β-(1,3)-glucan and ergosterol following the incubation period, fungi were harvested with a sieve and washed twice with distilled water, lyophilized and weighted in 15 mg aliquots. The β-(1,3)-glucan cell wall content was determined using the aniline blue assay, as described by Fernandes et al. [69]. The quantification of chitin was based on the measurement of glucosamine released by the acid hydrolysis of purified cell walls, also described previously by Fernandes et al. [69]. The quantification of ergosterol in the cell membrane was performed using a procedure described by Breivik and Owades [70]. Briefly, 15 mg of the lyophilized mycelia was incubated in a water bath at 85 °C for 1 h in 1.5 mL of a 25% KOH alcoholic solution. After cooling at room temperature, 0.5 mL of distilled water and 1.5 mL of *n*-heptane were added to the tubes and vortexed for 3 min. The organic phase containing ergosterol was stored at −20 °C for 24 h. The extracted ergosterol was diluted in 0.75 mL *n*-heptane and quantified by measuring the absorbance between 220 and 300 nm, as described previously [71].

### 4.6. Antioxidant Activity

The antioxidant activity of the extract was evaluated via three in vitro assays. (1) The capacity to inhibit the formation of thiobarbituric acid reactive substances (TBARS) was assessed using porcine brain tissues as oxidizable substrates, and the results were expressed as IC_50_ values (mg/mL), corresponding to the extract concentration providing 50% of antioxidant activity. (2) The oxidative hemolysis inhibition assay (OxHLIA) evaluated the extract capacity to protect sheep red blood cells (RBC) from the 2,2′-azobis(2-methylpropionamidine) dihydrochloride (AAPH)-induced oxidative hemolysis. The calculated IC_50_ values (mg/mL) translated the extract concentration required to keep 50% of the RBC population intact for a Δ*t* of 60 min. Both assays were previously described by [72] and Trolox (Sigma-Aldrich, St. Louis, MO, USA) was used as a positive control. (3) For the cellular antioxidant activity (CAA) assay, RAW264.7 cells were incubated with the extract and AAPH, using 2,7-dichlorofluorescein diacetate (DCFH-DA) as a fluorescent marker. DCF-DA is a compound that is easily oxidized by peroxide radicals to DCFH-DA, a fluorescent compound, once in the cell medium. The efficacy of antioxidant treatment was quantified as described previously [73]. Quercetin was used as a positive control, and DCFH and DMEM culture media were used as a negative control.

### 4.7. Anti-Inflammatory Activity

The anti-inflammatory potential of WcCEE extracts was evaluated on a RAW264.7 mouse macrophage cell line (DMSMZ—Leibniz Institute DSMZ—German Collection of Microorganisms and Cell Cultures GmbH), as described previously [74]. The extracts were tested for their capacity to inhibit lipopolysaccharide (LPS)-induced nitric oxide (NO) production on a murine macrophage cell line (RAW 264.7). Dexamethasone was used as the positive control and the samples without LPS were used as a negative control. The results were expressed as IC_50_ values (µg/mL), corresponding to the extract concentration responsible for 50% of NO production inhibition.

### 4.8. Cytotoxicity Activity

The cytotoxic potential of the WcCEE extract was evaluated on the following human tumor cell lines: AGS (gastric adenocarcinoma), CaCo2 (colorectal adenocarcinoma), MCF-7 (breast adenocarcinoma) and NCI-H460 (lung carcinoma); and on the non-tumor cell lines: Vero (African green monkey kidney) and PLP2 (primary porcine liver cell culture). All cell lines were maintained in an RPMI-1640 medium supplemented with 10% FBS, glutamine (2 mM), penicillin (100 U/mL) and streptomycin (100 mg/mL), except for Vero, kept in a DMEM medium supplemented with 10% FBS, glutamine and antibiotics. The culture flasks were incubated at 37 °C with 5% CO_2_, under a humid atmosphere. The cells were used only when reaching 70 to 80% confluence. Successive dilutions of the WcCEE extract were incubated with the cell suspension for 72 h, with the range of the final concentrations tested being 6.25–400 μg/mL. The 96-well microplates were incubated at 37 °C with 5% CO_2_, in a humid atmosphere, after checking for the adherence of the cells. All cell lines were tested at a concentration of 10 × 10^3^ cells/well, except for Vero, in which a density of 19 × 10^3^ cells/well was used. After the incubation period, the extract toxicity was screened using the sulforhodamine B assay, as described previously [71]. The results were expressed as the extract concentration (µg/mL) able to inhibit cell growth by 50%, i.e., GI_50_ values. Ellipticine was used as a positive control.

### 4.9. Statistical Analysis

Data analyses were performed using GraphPad Prism software (Version 8.0.1, Dr. Harvey Motulsky, San Diego, CA, USA). Results with *p* < 0.05 were considered statistically significant.

## 5. Conclusions

Several *Whitania* spp. have been described as having bioactive properties. With this work, the phytochemichal profile of a crude ethanolic extract of *W. chevalieri* (WcCEE), an endemic plant of Cape Verde archipelago, was reported for the first time. It can be concluded that WcCEE is an efficient inhibitor of dermatophyte growth, the fungi responsible for dermatophytoses, one of the most common skin infectious diseases. Moreover, for this antifungal/antidermatophytic activity, it should be noted that WcCEE synergizes with conventional antifungals used in human health to eradicate dermatophytoses, particularly terbinafine. WcCEE also proved to have antioxidant, anti-inflammatory and anticancer activities, without having cytotoxic effects in non-tumoral cell lines. Overall, it can be concluded that there is scientific evidence supporting the efficiency and safety of the *W. chevalieri* crude ethanolic extract prepared and used by Cape Verdeans in traditional medicine. This work also opens the perspective of studying bioactivities of individual compounds that we identified in the ethanolic crude extract of the whole *W. chevalieri* plant, envisaging novel therapeutic strategies.

## Figures and Tables

**Figure 1 plants-12-02502-f001:**
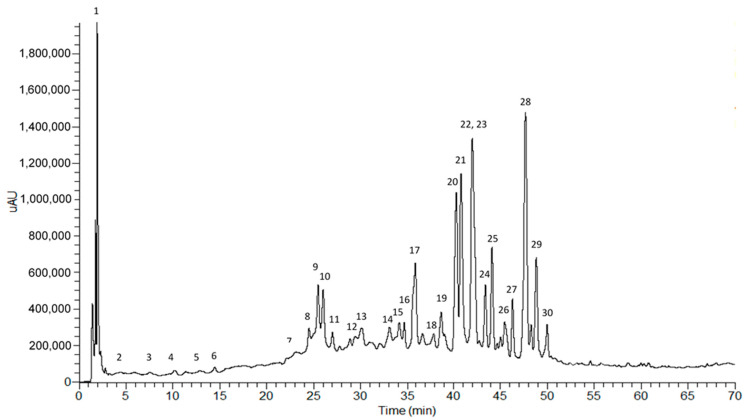
HPLC–PDA profile of WcCEE recorded between 200 and 600 nm. Peaks 1–30 correspond to the compounds listed in Table 1.

**Figure 2 plants-12-02502-f002:**
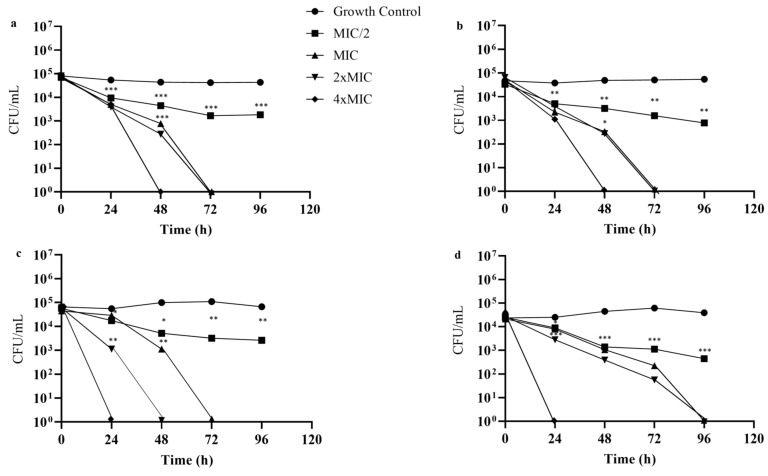
Time-kill curves of four selected strains of dermatophytes: (**a**) *T. soudanense* (CV3), (**b**) *T. rubrum* (CV55), (**c**) *T. interdigitale* (Esp14) and (**d**) *M. canis* (IMF35), based on the fungicidal action of WcCEE compared to the untreated control. Statistical analysis was conducted using a two-way ANOVA, followed by Turkey’s multiple comparison test to determine significance (***, *p* < 0.001; **, *p* < 0.05; *, *p <* 0.01) between the treated and untreated cells.

**Figure 3 plants-12-02502-f003:**
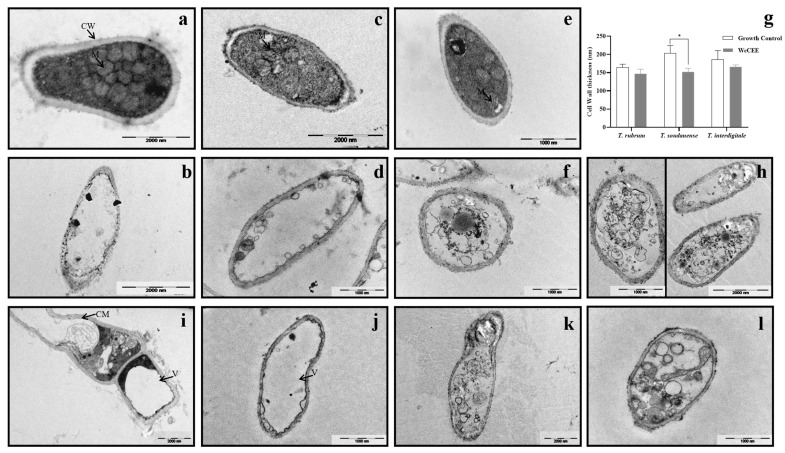
Representative TEM analysis of the ultrastructure of dermatophytes. The fungi *T. interdigitale* (Esp14) (**a**), *T. soudanense* (CV3) (**c**) and *T. rubrum* (CV55) (**e**) were grown under control conditions. *T. interdigitale* (**b**,**i**,**k**) and *T. soudanense* (**d**,**j**,**l**) were exposed to WcCEE; and *T. rubrum* was exposed to WcCEE (**f**) and itraconazole (**h**). The results of the cell wall thickness. measurement for *T. soudanense* (**g**) reflect the means SEMs of at least 20 random measurements (*, *p* > 0.05). The arrows indicate the cell wall (CW), plasma membrane (CM), mitochondria (M) and vacuole (V).

**Figure 4 plants-12-02502-f004:**
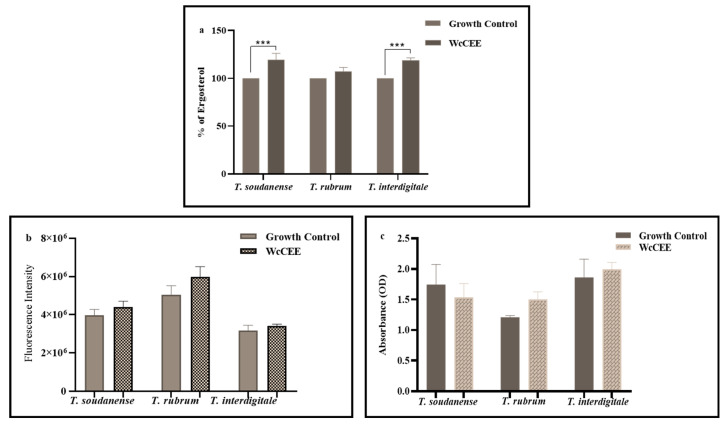
Effect of WcCEE on the fungal cell wall components, β-1,3-glucan and chitin, and the membrane ergosterol. Quantification of ergosterol (**a**), β-1,3-glucan (**b**) and chitin (**c**) when the fungi (*T. interdigitale* Esp14, *T. soudanense* CV3 and *T. rubrum* CV55) were grown in control conditions and with WcCEE (MIC value for each isolate). For ergosterol (**a**), the results are presented in percentage, considering the growth control as 100%. Results are the mean ± standard error of triplicates of three independent experiments (Turkey’s multiple comparison test), *p* > 0.05 (non-significant: ns); ***, *p* < 0.001.

**Figure 5 plants-12-02502-f005:**
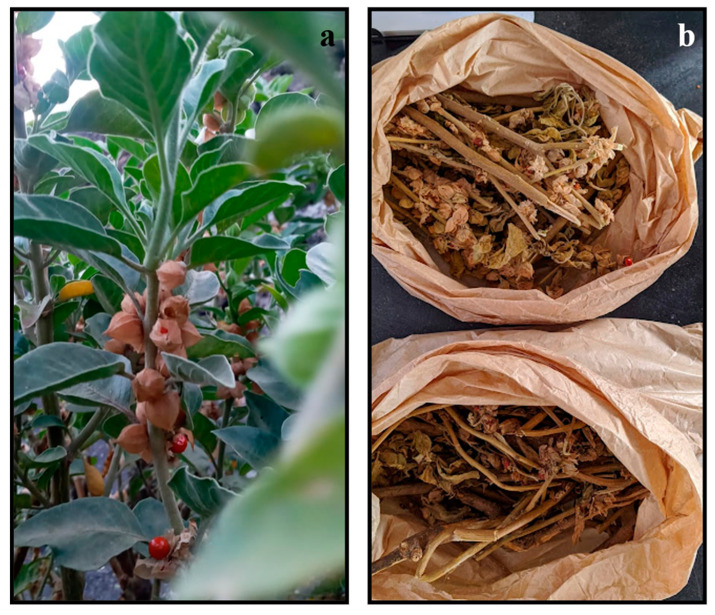
*W. chevalieri* harvested on Fogo Island (**a**) and after drying (**b**).

**Table 1 plants-12-02502-t001:** Chromatographic and spectral data of the compounds identified in WcCEE, obtained via HPLC–PDA–ESI–MSn.

ESI–MSn [*m*/*z* (Relative Abondance %)]						
Peak	R_t_ (min)	λ_max (nm)_	Precursor Ion [M–H]^−^ [M–H]^+^	MS^2^	MS^3^	Attempt to Identify
**1**	1.86	291	**191**	111 (100); 173 (33)	-	Citric acid (C_6_H_5_O_7_)
			**-**	-	-	
**2**	4.83	269	**-**	-	-	Phenylalanine (C_9_H_11_NO_2_)
			**166**	120 (100)	-	
**3**	7.88	291, 324	**353**	191(100); 179 (15); 173 (7); 161 (9)	-	5-*O*-Caffeoylquinic acid (C_16_H_18_O_9_)
			**-**	-	-	
**4**	8.40	291, 324	**367**	193(20); 191(10); 173(100); 135(9)	-	4-*O*-Feruloylquinic acid (C_17_H_20_O_9_)
			**-**	-	-	
**5**	13.32	318	**249**	-	-	Caffeoyl putrescine (C_13_H_18_N_2_O_3_)
			**251**	234 (100); 163 (5); 89 (10)	234(100), 163 (25)	
**6**	14.30	287, 320	**771**	609 (100)	301(100)	Quercetin-O-dihexosyl-deoxyhexoside (C_33_H_40_O_21_)
			**773**	627 (50); 611 (75); 665 (100)	303 (100)	
**7**	23.51	286, 320	**609**	301(100)	-	Quercetin-O-hexosyl-deoxyhexoside (C_27_H_30_O_16_)
			**611**	465(40); 303(100)	-	
**8**	25.10	290, 321	**-**	-	-	Feruloyl tyramine (C_18_H_19_O_4_N)
			**314**	177(100); 145(10); 117(2)	177(80); 145(100)	
**9**	25.32	254, 29 lsh, 338	**609**	609(50); 301(100)	301(100), 179(20); 151(15)	Quercetin-O-hexosyl-deoxyhexoside (C_27_H_30_O_16_)
			**611**	-	-	
**10**	26.45	282, 323	**-**	-	-	Bis(dihydrocaffeoyl)spermidine (C_25_H_35_N_3_O_6_)
			**474**	474 (100); 455 (15); 222 (60); 192 (5); 165 (10)	-	
**11**	27.12	284, 323	**-**	-	-	Methoxyferuloyltyramine (C_19_H_21_NO_5_)
			**344**	177(100); 145 (10)		
**12**	28.72	274, 350	**593**	285 (100)	-	Luteolin-O-hexosyl-deoxyhexoside (C_27_H_30_O_15_)
			**595**	449(40); 287(100)	-	
**13**	30.13	235	**501** 547	483 (95); 315 (100)	-	Baimantuoluoside J (C_30_H_44_O_9_)
			**-**	-	-	
**14**	33.10	230	**-**	-	-	Withanoside II isomer (C_40_H_62_O_16_)
			**817** [M+NH_4_]^+^	799 (100); 637(40); 475(45)	475 (100)	
**15**	34.12	230	**-**	-	-	Withanoside II isomer (C_40_H_62_O_16_)
			**817** [M+NH_4_]^+^	799 (100); 637(30); 475(40)	475 (100)	
**16**	34.67	-	**-**	-	-	Grossamide (C_36_H_37_N_2_O_8_)
			**625**	488 (10); 462 (100); 351 (5); 325 (30)	325 (100)	
**17**	35.92	231	**549** [M+HCOO]^−^	503 (100)	361(100); 377(80); 467(75); 343(60)	Withanolide S (C_28_H_40_O_8_)
			**522** [M+NH_4_]^+^	505 (3); 488 (100); 319 (95); 169 (15)	-	
**18**	36.77	228	**973**	955 (100)	831(831)	Ashwagandhanolide (C_56_H_78_O_12_S)
			975 **504**	469 (100); 301 (30); 283 (20); 265 (50)	451(100)	
**19**	38.98	230	**990**	943 (100)	781(100)	Withanolide sulfoxide (C_56_H_78_O_13_S)
			996 [M+HCOO]^−^ **504**	469 (100)	-	
**20**	40.21	234	**827**	781 (100)	781(100); 764(10); 619(15)	Withanoside IV (C_40_H_63_O_15_)
			**783** 800 [M+NH_4_]^+^	621 (40); 459 (100); 441 (80): 423 (40); 405 (20)	441(100); 423(30); 269 (10)	
**21**	40.89	234	**827** [M+HCOO]^−^	781 (100)	781(100); 764(10); 619(15)	Withanoside VI (C_40_H_63_O_15_)
			**783** 800 [M+NH_4_]^+^	621 (40); 459 (100); 441(80): 423(40); 405(20)	441(100); 423(30) 169(20)	
**22**	41.93	233	**515** [M+HCOO]^−^	469(100)	451(100); 433(10); 327(60)	δ-lactol withanolide (type A) (C_28_H_38_O_6_)
			**453** 471 [M+H]^+^ 488 [M+NH_4_]^+^ 493 [M+Na]^+^ 941 [2M+H]^+^ 958 [2M+NH_4_]^+^	471(5); 435(100); 417(40); 399(15); 325 (10)	417(100); 399(50); 323(70)	
**23**	42.38	232	**665** [M+HCOO]^−^	619(100)	439(100)	Coagulin Q/physagulin D (C_34_H_53_O_10_)
			**621**	459(100); 441(95); 423(40); 405(20)	441(100); 423(40); 405(20)	
**24**	43.43	230	**515** [M+HCOO]^−^	469(100)	451(100); 433(10) 327(80)	δ-lactol withanolide (type A) (C_28_H_38_O_6_)
			**488** [M+NH_4_]^+^ 453 493 [M+Na]^+^ 941 [2M+H]^+^ 958 [2M+NH_4_]^+^	471(5); 435(100); 417(40); 399(15)	417(100); 399(50);	
**25**	44.19	231	**469** 515 [M+HCOO]^−^	343 (100)	325(55); 307(40); 299(100); 273(10)	δ-lactone withanolide (type A) (C_28_H_38_O_6_)
			**453** 493 [M+Na]^+^ 941 [2M+H]^+^ 958 [2M+NH_4_]^+^	435(75); 417(100); 399(40); 267(15)	399(100); 381(15); 289(20); 211(15)	
**26**	44.56	228	**753**	591(100)	429(100)	Withanamide B/C (C_38_H_63_N_2_O_13_)
			755	593 (100)	431 (100)	
**27**	46.32	229	**515** [M+HCOO]^−^	469 (100)	451 (90); 345 (100)	δ-lactol withanolide (type A) (C_28_H_38_O_6_)
			**493** [M+Na]^+^ 453 488 [M+NH_4_]^+^ 941 [2M+H]^+^ 958 [2M+NH_4_]^+^	453(95); 285(100); 169(70)	267(100); 249(30); 201(20); 157(30)	
**28**	47.67	235	**-**	-	-	Withanoside II aglycone (C_28_H_43_O_6_)
			**475**	475(100); 457(5); 421 (5)	-	
**29**	48.60	228	**499** [M+HCOO]^−^	453 (100); 451 (20)	435 (100); 409 (75)	Withacoagulin isomer (C_28_H_38_O_5_)
			**472** [M+NH_4_]^+^ 477 [M+Na]^+^ 926 [2M+NH_4_]^+^	453 (30); 437 (75); 267 (20); 169 (15)	-	
**30**	49.21	228	**499** [M+HCOO]^−^	453 (100); 451 (25)	435 (100); 409 (65)	Withacoagulin isomer (C_28_H_38_O_5_)
			**472** [M+NH_4_]^+^ 455 [M+H]^+^ 477 [M+Na]^+^ 926 [2M+NH_4_]^+^	437 (20); 267 (5); 169 (2)	-	

**Table 2 plants-12-02502-t002:** Susceptibility of the clinical isolates of dermatophytes: MIC and MFC values of the ethanolic extract of *W. chevalieri* and MIC values for itraconazole (IT), terbinafine (TB) and griseofulvin (GF).

Strain ID	Species	MIC					
		WcCEE (mg/mL)			IT (µg/mL)	TB (µg/mL)	GR (µg/mL)
		MIC	MFC	MFC/MIC			
CV3	*T. soudanense*	3.12	3.12	1	0.031	0.06	4
CV4	*T. soudanense*	6.25	6.25	1	0.062	0.03	2
CV8	*T. soudanense*	6.25	6.25	1	0.015	0.13	1
CV10	*T. soudanense*	6.25	6.25	1	0.062	0.007	1
CV11	*T. soudanense*	6.25	6.25	1	0.062	0.007	1
CV12	*T. soudanense*	1.56	1.56	1	0.062	0.015	2
CV15	*T. soudanense*	6.25	6.25	1	0.062	0.015	1
CV20	*T. soudanense*	1.56	1.56	1	0.062	0.015	4
CV24	*T. soudanense*	3.12	3.12	1	0.125	0.015	8
CV30	*T. soudanense*	6.25	6.25	1	0.031	0.007	0.5
CV42	*T. soudanense*	6.25	6.25	1	0.003	0.007	1
CV45	*T. soudanense*	6.25	6.25	1	0.031	0.007	4
CV47	*T. soudanense*	6.25	6.25	1	0.031	0.007	4
CV50	*T. soudanense*	3.12	3.12	1	0.031	0.007	0.5
CV52	*T. soudanense*	1.56	3.12	1	0.031	0.007	2
CV 54	*T. soudanense*	3.12	3.12	1	0.015	0.003	4
CV58	*T. soudanense*	6.25	6.25	1	0.125	0.015	2
CV60	*T. soudanense*	6.25	6.25	1	0.015	0.125	2
Esp16	*T. rubrum*	6.25	6.25	1	0.062	0.06	1
Esp32	*T. rubrum*	6.25	6.25	1	0.125	0.02	1
IMF-29	*T. rubrum*	3.12	6.25	2	0.25	0.015	1
CV55	*T. rubrum*	6.25	6.25	1	0.007	0.03	4
Esp14	*T. interdigitale*	6.25	6.25	1	0.031	0.06	0.5
IMF-28	*T. interdigitale*	3.12	6.25	2	0.125	0.13	2
IMF-35	*M. canis*	25	25	1	0.25	0.5	2
PT01	*M. canis*	25	25	1	0.25	0.5	2

**Table 3 plants-12-02502-t003:** MIC of *W. chevalieri* crude ethanolic extract (WcCEE) in combination with itraconazole (IT), terbinafine (TB) and griseofulvin (GR).

		MIC (mg/mL) and (μg/mL)				MIC of WcCEE (mg/mL) + IT (μg/mL)				MIC of WcCEE (mg/mL) + TB (μg/mL)				MIC of WcCEE (mg/mL) + GR (μg/mL)			
Strain ID	Species	WcCEE	IT	TB	GR	WcCEE	IT	FICI	Observ.	WcCEE	TB	FICI	Observ.	WcCEE	GR	FICI	Observ.
CV8	*T. soudanense*	6.25	0.015	0.13	1	0.097	0.007	0.48	SYN. *	0.097	0.001	0.024	SYN.	1.56	0.125	0.375	SYN.
CV4	*T. soudanense*	6.25	0.062	0.03	2	1.56	0.031	0.75	ADD. **	0.781	0.007	0.351	SYN.	0.781	1	0.625	ADD.
CV3	*T. soudanense*	3.12	0.031	0.06	4	0.097	0.007	0.26	SYN.	0.097	0.015	0.273	SYN.	0.39	1	0.375	SYN.
Esp14	*T. interdigitale*	6.25	0.031	0.06	0.5	0.097	0.007	0.24	SYN.	0.097	0.015	0.257	SYN.	3.12	0.125	0.749	ADD.
IMF28	*T. interdigitale*	3.12	0.125	0.13	2	1.56	0.062	1	ADD.	0.781	0.031	0.498	SYN.	1.56	1	1	ADD.
Esp32	*T. rubrum*	6.25	0.125	0.02	1	0.097	0.062	0.51	ADD.	0.097	0.003	0.216	SYN.	0.097	0.5	0.516	ADD.
Esp16	*T. rubrum*	6.25	0.062	0.06	1	0.195	0.062	1.03	ADD.	0.39	0.015	0.304	SYN.	1.56	1	1.25	ADD.
CV55	*T. rubrum*	6.25	0.007	0.03	4	0.097	0.003	0.44	SYN.	0.097	0.007	0.241	SYN.	3.12	0.5	0.624	ADD.
IMF35	*M. canis*	25	0.25	0.5	2	25	0.25	2	ADD.	6.25	0.062	0.374	SYN.	25	2	2	ADD.
PT01	*M. canis*	25	0.25	0.5	2	6.25	0.25	1.25	ADD.	0.097	0.125	0.254	SYN.	25	2	2	ADD.

* SYN.—Synergy between antifungal and WcCEE; ** ADD.—Additive effect.

**Table 4 plants-12-02502-t004:** Development of resistance. MIC values following repeated exposure to WcCEE.

MIC (mg/mL)				
Isolate	Initial	5 Passages	10 Passages	15 Passages
*T. rubrum* CV55	6.25	6.25	6.25	6.25
*T. interdigitale* Esp14	6.25	6.25	6.25	12.5 *
*T. soudanense* CV3	6.25	6.25	6.25	6.25

* From the 13th passage.

**Table 5 plants-12-02502-t005:** Antioxidant, anti-inflammatory and cytotoxic activity of WcCEE.

	WcCEE	Positive Control
Antioxidant activity		
TBARS (IC_50_, mg/mL)	2.1 ± 0.2	0.0091 ± 0.0003
OxHLIA (IC_50_, mg/mL)	0.49 ± 0.03	0.0218 ± 0.0003
CAA (% inhibition)	60 ± 5	95 ± 5
Anti-inflammatory activity (IC_50_, µg/mL)		
NO production inhibition	7 ± 1	6.3±0.4
Cytotoxic activity (GI_50_, µg/mL)		
AGS	47 ± 4	1.20 ± 0.03
CaCo2	63 ± 4	1.20 ± 0.02
MCF-7	27 ± 2	1.00 ± 0.02
NCI-H460	19 ± 1	1.20± 0.02
Vero	>400	1.40 ± 0.06
PLP2	>400	1.4 ± 0.1

Positive controls: Trolox (TBARS and OxHLIA), quercetin (CAA), ellipticine (cytotoxicity) and dexamethasone (anti-inflammatory activity).

**Table 6 plants-12-02502-t006:** Clinical isolates of dermatophytes used in this study.

Code	Species	Anatomical Site	Origin
CV3	*T. soudanense*	Body	Cape Verde
CV4	*T. soudanense*	Hand	Cape Verde
CV8	*T. soudanense*	Nail	Cape Verde
CV10	*T. soudanense*	Leg	Cape Verde
CV11	*T. soudanense*	Body	Cape Verde
CV12	*T. soudanense*	Arm	Cape Verde
CV15	*T. soudanense*	Face	Cape Verde
CV20	*T. soudanense*	Body	Cape Verde
CV24	*T. soudanense*	Leg	Cape Verde
CV30	*T. soudanense*	Body	Cape Verde
CV42	*T. soudanense*	Arm	Cape Verde
CV45	*T. soudanense*	Body	Cape Verde
CV47	*T. soudanense*	Body	Cape Verde
CV50	*T. soudanense*	Toenail	Cape Verde
CV52	*T. soudanense*	Hand	Cape Verde
CV54	*T. soudanense*	Nail	Cape Verde
CV58	*T. soudanense*	Face	Cape Verde
CV60	*T. soudanense*	Scalp	Cape Verde
Esp16	*T. rubrum*	Nail	Spain
Esp32	*T. rubrum*	Nail	Spain
IMF-29	*T. rubrum*	Nk	Portugal
CV55	*T. rubrum*	Nail	Cape Verde
Esp14	*T. interdigitale*	Nail	Spain
IMF-28	*T. interdigitale*	Nk	Portugal
IMF-35	*M. canis*	Nk	Portugal
PT01	*M. canis*	Arm	Portugal

Nk—not known.

## Data Availability

Data will be available upon request.

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
