# Peer review of "The Chemical Profile, and Antidermatophytic, Anti-Inflammatory, Antioxidant and Antitumor Activities of Withania chevalieri A.E. Gonç. Ethanolic Extract"

_plants, 2023, doi:10.3390/plants12132502_

Round 1

Reviewer 1 Report

-The manuscript should be checked by a native English speaker and be corrected for grammatical errors.

- The title need to reform, I suggest : Chemical Profile and Antidermatophytic, Antioxidant and Antitumor Activities of ethanolic extract of the Cape Verde Endemic Medicinal Plant Withania chevalieri A.E. Gonç.

- The Authors must state the date of plant material collected, provide the GPS. Indeed, the nature and the quantity of the constituent components of the extracts of the same plant vary according to these parameters.

- Page 15, line 413, the Authors stated that «The whole plant was separated into the different parts (roots, stem, fruits and leaves) and ground into powder »: are the different parts of the plant separately ground (there is an ambiguity, you have to be precise!)? if not, why did the authors not study the extracts of the different parts of the plant separately? Indeed, this operation will make the search for the active compounds of the plant more precise.

- Page 15 lines  414-415, the Authors stated that « The extraction method was the closest to the traditional process, using the method presented by Nefzi et al. [48], with modifications : what is the modification apported by authors to this method ?

- Why did the Authors not attempt to carry out separate tests for each of the main components (or the fraction of the ethanolic extract) of Withania cheva-lieri A.E. Gonç.? The research does not show which component(s) (or fraction of the ethanolic extracts) of Withania chevalieri A.E. Gonç. is biologically active?

- Pages 13-14: the part from line 292 to line 306 “Dermatophytosis ……………………..and anticancer bioactivities) should be put in the introduction and not in the discussion part.

- No conclusion !

- Source of literature used, the number of bibliographical references should be reduced: more than a hundred references. This is not a literature review!! Moreover, more than 25 references are more than 10 years old (1998-2013).

Author Response

Dear Reviewer

On behalf of all the authors I deeply acknowledge the comments and the questions that made us revise thoroughly the entire manuscript contributing to increase the quality and readability of the MS. Besides the specific changes made in response to your queries (see Response point-by-point below), several general aspects were modified, also in response to the other reviewers. The manuscript was carefully edited in what regards English language and grammar and corrected accordingly; throughout the MS, typing errors and imprecisions were corrected. The excessive number of references was shorten from 102 to 71. In this resubmission we also include a Graphical abstract that we believe will allow the reader of PLANTS to gain interest in our work.

Sincerely,

Teresa Gonçalves

RESPONSE TO REVIEWER #1 POINT-BY-POINT

Comments and Suggestions for Authors

-The manuscript should be checked by a native English speaker and be corrected for grammatical errors.

This was one of our most important concerns when making the revision of the original manuscript. The entire manuscript was revised and modified to  improve english language and grammar. 

- The title need to reform, I suggest : Chemical Profile and Antidermatophytic, Antioxidant and Antitumor Activities of ethanolic extract of the Cape Verde Endemic Medicinal Plant Withania chevalieri A.E. Gonç.

The title was modified as suggested by the reviewer to:

Chemical Profile and Antidermatophytic, Antioxidant, Anti-inflammatory and Antitumor Activities of Ethanolic Extract of the Cape Verde Endemic Medicinal Plant Withania chevalieri A.E. Gonç.

 In the keywords list we also added “anti-inflammatory”

- The Authors must state the date of plant material collected, provide the GPS. Indeed, the nature and the quantity of the constituent components of the extracts of the same plant vary according to these parameters.

The plants were collected between August and September 2021 and the GPS coordinates where the plants were collected are

Latitude: 14 56' 00'';  
Longitude: -24 22' 00''

Altitude:700 m

This information was added in the MS text, as follows:

(Line 408-410 clean MS; 492-494 marked MS): “The plants (Figure 5a) were collected in the volcanic area of Fogo Island, at Chã das Caldeiras (Latitude: 14 56' 00''; Longitude: -24 22' 00''; Altitude: 700 m), between the 29th August and the 6th of September 2021.”

- Page 15, line 413, the Authors stated that «The whole plant was separated into the different parts (roots, stem, fruits and leaves) and ground into powder »: are the different parts of the plant separately ground (there is an ambiguity, you have to be precise!)? if not, why did the authors not study the extracts of the different parts of the plant separately? Indeed, this operation will make the search for the active compounds of the plant more precise.

The authors are thankful for this question of the reviewer. In fact, the description of the methodology was misleading and not correct. Initially, when this work was initiated, the methodology followed was to separate each part of the plant, ground, and make separate extracts from each part of the plant. These were tested separately in what regards (only) the antifungal effect. By then, the conclusion was that an extract obtained from the whole plant had better antifungal activity than each part separately. So, it was decided to use the whole plant (including all the parts roots, stem, fruits, flower and leaves. This result suggests a synergy between compounds from different parts of the plant. This  is not a novelty, and references are found in the scientific literature describing these phenomena in other plants and how the separation of parts/components decreases the bioactivity (doi: 10.1186/1475-2875-10-S1-S4; https://doi.org/10.1039/C9NP00011A).

Nevertheless, we agree with the reviewer that with this methodology is more difficult to select the precise active compounds. On the other hand, this is also interesting because the overall bioactivity of the extract can result from different compounds acting in synergy.

To increase the clarity of how the extract was obtained, section 4.1 was corrected as follows:

(Line 412-419 clean MS; 496-507 marked MS)  

“… After being washed with water to remove dust, the plants were dried (Figure 5b) outdoors for 14 days and more 5 days in an incubator at 50 °C, to remove residual moisture. The whole plant (roots, stem, fruits, flower and leaves) was ground into powder. The extraction procedure was the closest to the traditional process. To initiate the ethanolic extraction, 10 g of the powder were mixed with 100 mL of 70% ethanol, and incubated at 25 ºC, during 4 days in a shaker at 150 rpm. This suspension was filtered using gauze and a Buchner funnel, followed by filtration in a common funnel with gauze, and evaporated at 60 ºC, for 24 h, …”

- Page 15 lines  414-415, the Authors stated that « The extraction method was the closest to the traditional process, using the method presented by Nefzi et al. [48], with modifications : what is the modification apported by authors to this method ?

This comment by the reviewer lead us to revise carefully the methodology described by Neftzi and collaborators in 2016 (doi: 10.4172/1948-5948.1000277) and in fact this is different from that used by Cape Verdeans. The reference was removed and the methodology was that described in detail in section 4.1 (Line 412-419 clean MS; 496-507 marked MS), as stated in the response to the previous comment/question.

- Why did the Authors not attempt to carry out separate tests for each of the main components (or the fraction of the ethanolic extract) of Withania chevalieri A.E. Gonç.? The research does not show which component(s) (or fraction of the ethanolic extracts) of Withania chevalieri A.E. Gonç. is biologically active?

As stated in the response to the second comment/question of this reviewer, we begin by testing ethanolic extracts of each part of the plant, separately, only in what regards to antidermatophytic effect. These preliminary assays showed that the extract of the whole pant extract had higher activity than the extracts obtained from the different parts of the plant. Based on this, it was decided to work with the whole plant extract, which mimics the methodology used in ethnomedicine. This a first approach to the study of the bioactivities of a crude ethanolic extract of Whitania chevalieri.

Nevertheless, the authors acknowledge the reviewer’ suggestion. The knowledge about isolated compounds or different fractions (for example, using serial extractions with different solvents) is always enlightening of the potential use of a more purified natural extract. This will be taken in consideration for future developments of this project.  

- Pages 13-14: the part from line 292 to line 306 “Dermatophytosis ……………………..and anticancer bioactivities) should be put in the introduction and not in the discussion part.

The authors agree with the reviewer and this paragraph was removed from the beginning of the Discussion section. Instead we added the following:

(Line 322-324 clean MS; 357-359 marked MS) “In this work we characterized for the first time the composition of an ethanolic extract of W. chevalieri, an endemic plant of Cape Verde used in traditional medicine, and evaluated several bioactivities, such as, antidermatophytic, antioxidant, an-ti-inflammatory and anticancer. ”     

- No conclusion !

The section  Conclusions was added to the manuscript:

(Line 601-612 clean MS; 695-706 marked MS)

“5. Conclusions

Several Whitania spp. have been described as having bioactive properties. With this work it is reported for the first time the phytochemichal profile of a crude ethanolic extract of W. chevalieri (WcCEE), an endemic plant of Cape Verde archipelago. It can be concluded that WcCEE is an efficient inhibitor of the growth of dermatophytes, the fungi responsible for dermatophytoses, one of the most common skin infectious diseases. Moreover, to this antifungal/antidermatophytic activity, it adds that WcCEE synergizes with conventional antifungals used in human health to eradicate dermatophytoses, in particular with terbinafine. WcCEE also proved to be antioxidant, anti-inflammatory and anticancer, without having cytotoxic effects in non-tumoral cell lines. Overall, it can be concluded that there are scientific evidences supporting the efficiency and safeness of the W. chevalieri crude ethanolic extract prepared and used by Cape Verdeans in traditional medicine.”

- Source of literature used, the number of bibliographical references should be reduced: more than a hundred references. This is not a literature review!! Moreover, more than 25 references are more than 10 years old (1998-2013).

The number of references was reduced from 102 to 71, from which only 22 were before 2013.

An attempt was made to decrease the number of references, to keep the most updated, but some more than 10 years had to be kept. The reason is that some are unique and cannot be replaced for more recent studies/reviews.

Reviewer 2 Report

Dear authors 

find attached 

Author Response

Dear Reviewer

On behalf of all the authors I deeply acknowledge the comments and the questions that made us revise thoroughly the entire manuscript contributing to increase the quality and readability of the MS. Besides the specific changes made in response to your queries (see Response point-by-point below), several general aspects were modified, some in which in response to the other reviewers. The manuscript was carefully edited in what regards English language and grammar and corrected accordingly; throughout the MS typing errors and imprecisions were corrected. The excessive number of references was shorten from 102 to 71. In this resubmission we also include a Graphical abstract that we believe will allow the reader of PLANTS to gain interest in our work.

Sincerely,

Teresa Gonçalves

RESPONSE TO REVIEWER POINT-BY-POINT

Comments

  1. 1-  Abstract:
    The authors should mentioned that (Withania chevalieri) is considered a medical plant, ornamentals or other for their used in the first lines ( 20 and 21).

Based on this comment we added the  classification of Whitania chevalieri as a medicinal plant, as follows:

(Line 21-22 clean MS; 21-23 marked MS)

“Withania chevalieri, endogenous from Cape Verde, is considered a medicinal plant, used in ethno-medicine with a large spectrum of applications, such as to treat skin fungal infections caused by dermatophytes. ,…”.

  1. 2-  Introduction:

In general, the introduction is very short need to add more references to give the readers clear idea about the background of this article. The comments as below:

  • -  Line 50 to 66 more references need about Withania chevalieri and their important varieties at least 2 or more will be good. This can show the readers the important of these plant varieties and their medical , pharmacy uses and other uses.

As far as the authors could access published information, there are no varieties (cultivars) of Withania chevalieri described. For another species of the Whitania genus,  Whitania somnifera, also known as Ashwagandha, several cultivars/varieties were described (https://doi.org/10.1016/j.sajb.2022.10.039). Overall the information about Withania chevalieri is scarce.

-  Again more references needed to be add to Line 50 to 60 about previous anticancer researches and their application on human and other Organisms 

The reviewer is probably referring to the fact that for Whitania somnifera there are several studies demonstrating the anticancer activity of extracts of this plant. For Whitania chevalieri there are no studies on its bioactivity, including its activity against cancer cells and our study is the first showing this bioactivity of W. chevalieri. Nevertheless, two more references were added:

 Palliyaguru DL, Singh SV, Kensler TW. Withania somnifera: From prevention to treatment of cancer. Mol Nutr Food Res. 2016 Jun;60(6):1342-53. doi: 10.1002/mnfr.201500756. 

Kashyap, V.K.; Peasah-Darkwah, G.; Dhasmana, A.; Jaggi, M.; Yallapu, M.M.; Chauhan, S.C. Withania somnifera: Progress towards a Pharmaceutical Agent for Immunomodulation and Cancer Therapeutics. Pharmaceutics 202214, 611. https://doi.org/10.3390/pharmaceutics14030611

  • -  Some information about HPLC-MS the their uses with extract the phenolic phenolic acids, flavonoids and terpenes. Please add more reference.

The information regarding HPLC-MS was described in more references were added to the manuscript.

  • -  Line 68 to 70 the authors should explain for the aim if they can applied their methods on different organisms that will make the manuscript more value.

Based on this comment of  the reviewer we reformulated the rationale and the objective (Lines 64-72 clean MS; 75-87 marked MS), as follows:

“Based on ethnopharmacological information, homemade formulations of W. chevalieri, including tinctures (dried plant soaked in 70% ethanol) or ointments (powder of plant mixed with sterile petroleum jelly), are widely used as antimicrobials, particularly in fungal skin infections such as dermatophytoses, and in skin ulcers due to infections by several bacteria. These formulations are also used as anti-inflammatory and as anticancer. However, there are no scientific data on the biological activities and chemical composition of these extracts. Therefore, the objectives of this work were, for the first time, to chemically characterize W. chevalieri ethanolic extract (WcCEE) and to make an evaluation of its biological properties supporting the traditional uses of this plant.”

  1. 3-  Results:
    • -  For the Table 1, there is a huge numbers and information were added in the table and that resulting to confuse the readers and no one will understand the presenting information. I understand that the authors need to present all their data but if they can delete some unimportant numbers or they can divided the table into two tables. Also, they can used some figures to present their results that will be good.

In our opinion, the data presented in the table, which include the retention time,  uv maxima and mass spectrometry fragments, are indispensable for the identification of the compounds in the extract. To improve the understanding of Table 1, we added a brief explanation and interpretation of these data.

  • -  2.1.1to 2.1.5, some abbreviations are not clear, so the authors should write the abbreviations in full for the first time and then they can use them.

The following abbreviations were written in full:

 “high performance liquid chromatography coupled with a photodiode array detector and a mass spectrometry detector with a electrospray interface (HPLC-PDA-ESI-MSn )”

 “ultra violet/visible (UV/vis) spectra”

  • -  For table 3, can be used as Landscape design for the paper , but this can be organized as journal style and editor decision. The reason is because the information there is too small and not clear.

The authors acknowledge the reviewer comment. In fact Table 3 was difficult to read. We changed it to Landscape.

  1. 4-  Discussion

- Lines 308- 320, this part is weak so need more explanations with good scientific reasons. One or more reference need to add.

This paragraph was modified accordingly (Lines 325-338 clean MS; 386-390 marked MS:

“WcCEE was characterized by organic acids (in particular citric acid), essential amino acids (phenylalanine), phenolic acids (5-O-caffeoylquinic acid and 4-O-feruloylquinic acid) flavonoids (quercetin and luteolin), phenolamides (bis(dihydro caffeoyl)spermidine, caffeoyl putrescine, feruloyl tyramine, and methoxyferuloyltyramine), a large group of terpenes called withanolides, and other active compounds, all known to be important compounds with beneficial medicinal properties [10,30,31]. Related to described com-pounds found in other Withania species, recent studies, of different extracts, analyzed by different methods revealed the withanolides as being the major constituents of W. somnifera along with different classes of withanosides (4-OH and 5,6-epoxy withanolides (withaferin A-like steroids) and 5-OH and 6,7-epoxy withanolides (withanolides A-like steroids)[32], lipids, sugars, amino acids, organic acids, phenolic compounds, flavonoids and many other secondary metabolites with broad spectrum of activity, as recently re-viewed [33]. These data are in agreement with the results obtained in our study with W. chevalieri.”

 Some references were included in this text to demonstrate the effects and the composition of other Whitania spp.

5- Materials and Methods

  • -  4.3. Chemical Characterization of the Crude Ethanolic Extract/ information about the HPLC is not complete such as (Agilent Technologies Inc., Santa Clara, CA, USA) with serial number.
  • -  Again, the information about formic acid need to be add as above information.

The information was completed as follows:

(Line 433-444 clean MS; 521-534 marked MS)

“4.3. Chemical Characterization of the Crude Ethanolic Extract

The chemical characterization of the WcCEE was performed using a Liquid chro-matograph with a photodiode spectrophotometer-PDA detector (Thermo Finnigan Sur-veyor, San Diego, CA, USA) interfaced with a linear ion trap mass spectrometer (LIT-MS) (LTQ XL, Thermo Scientific, Waltham, MA, USA). HPLC (Finnigan Surveyor, THERMO) coupled to a Diode Array Spectrometer (Finnigan Surveyor, THERMO) and a Linear Ion Trap Mass Spectrometer (LIT-MS) (LTQ XL, ThermoScientific). The LC column was a Waters Spherisorb ODS2 (3 µm, 150x2.1 mm) (Waters Corporation, Milford, Massachu-setts, USA) and separation was carried at 20 ºC. 20 μL of the dissolved extract (2.65 mg/mL in methanol 50%) was injected and the elution was performed with a mobile phase con-sisting of 2% aqueous formic acid (solvent A) and methanol (solvent B). Formic acid and methanol (HPLC grade) were purchased from Merck (Lisbon, Portugal). The gradient profile used was 0-70 min, 20-100% B. The flow rate was 200 µL/min. The first detection was made in the diode array spectrophotometer between 220 and 650 nm.”

6- Conclusions: must be at the manuscript because the discussion is long and complex. This section should be concentrate on the high significant and interesting results only.

The following conclusion section was added (Line 601-612 clean MS; 695-706 marked MS):

  1. Conclusions

Several Whitania spp. have been described as having bioactive properties. With this work it is reported for the first time the phytochemichal profile of a crude ethanolic extract of W. chevalieri (WcCEE), an endemic plant of Cape Verde archipelago. It can be concluded that WcCEE is an efficient inhibitor of the growth of dermatophytes, the fungi responsible for dermatophytoses, one of the most common skin infectious diseases. Moreover, to this antifungal/antidermatophytic activity, it adds that WcCEE synergizes with conventional antifungals used in human health to eradicate dermatophytoses, in particular with terbinafine. WcCEE also proved to be antioxidant, anti-inflammatory and anticancer, without having cytotoxic effects in non-tumoral cell lines. Overall, it can be concluded that there are scientific evidences supporting the efficiency and safeness of the W. chevalieri crude ethanolic extract prepared and used by Cape Verdeans in traditional medicine.

7- References

  • -  The first 3 references should be deleted because these are not references just Instructions how to write the references.

Removed

  • -  The references are more than need it, so some of these references must delete such as Table 1 (Attempt to Identify) one reference is enough for each chemical structure and the rest over need.

The references were deleted from Table 1.

The number of references was reduced from 102 to 71, from which only 22 were before 2013.

  • -  All in all, the references need to be organized again in text or in references list according to their order in the manuscript.

The order of the reference list was carefully revised and corrected.

Reviewer 3 Report

Dear Authors,

The present study evaluates the antidermatophytic activity of a W. chevalieri crude ethanolic extract also and assesses its in vitro antioxidant, anti-inflammatory and cytotoxic effects. The research subject is interesting and brings scientific important data in the field. Some changes of the manuscript should nevertheless be performed in order to improve its quality. Following specific changes should thus be performed:

Major changes

Abstract: It should follow the structure of the manuscript, having the same structure.

Introduction: In this section, authors need to compare the purposes of the present study with similar studies existing in literature. These studies need an adequate presentation. Afterwards, authors need to clarify what their study brings in novelty. It is very important to state what exactly you bring in novelty in order to express your originality. The purpose of the study needs to be found in the last paragraph and be clearer. In fact, the whole section is unadequately organized: too many general informations in the first paragraph and too little specific ones in the second one. The specific part needs to be detailed. You need to offer informations on the state of the art of concepts that are included in your study. Please add further information and justifications and modify accordingly.

Discussions: The first paragraph of this section is not adequately placed, it is more like an Introduction. Moreover, novelty and originality of your study is essential to be emphasized in this section once again. You need to emphasize this in terms of results, not purposes, as the Introduction should. You need to connect more the different concepts which are tested in this study.

Materials and Methods: You do not offer references for neither of the assays. Are all methods completely new? Because, if they are, you need to offer them a different approach.

Conclusions: This section needs to be added, as Results and Discussion section is quite long. Please add perspectives of your study.

References should follow the recommendations of the journal.

English needs serious revisions, if possible please review the whole manuscript, preferably by a native english speaker.

All these suggested changes should be performed in order to bring further improvements to the manuscript. 

English needs serious revisions, if possible please review the whole manuscript, preferably by a native English speaker.

Author Response

Dear Reviewer

On behalf of all the authors I deeply acknowledge the comments and the questions that made us revise thoroughly the entire manuscript contributing to increase the quality and readability of the MS. Besides the specific changes made in response to your queries (see Response point-by-point below), several general aspects were modified, together with the modifications made as requested by other reviewers. The manuscript was carefully edited in what regards English language and grammar and corrected accordingly; throughout the MS typing errors and imprecisions were corrected. The excessive number of references was shorten from 102 to 71. In this resubmission we also include a Graphical abstract that we believe will allow the reader of PLANTS to gain interest in our work.

Sincerely,

Teresa Gonçalves

RESPONSE TO REVIEWER POINT-BY-POINT

Comments and Suggestions for Authors

Dear Authors,

The present study evaluates the antidermatophytic activity of a W. chevalieri crude ethanolic extract also and assesses its in vitro antioxidant, anti-inflammatory and cytotoxic effects. The research subject is interesting and brings scientific important data in the field. Some changes of the manuscript should nevertheless be performed in order to improve its quality. Following specific changes should thus be performed:

Major changes 

Abstract: It should follow the structure of the manuscript, having the same structure.

The authors acknowledge the reviewer comment. The abstract was written to follow that scheme, although the titles of the subsection were not introduced, as indication in the Journal’s template. Nevertheless, your comment allowed a revision of the abstract and modifications were introduced to cope with the organization needed.

Please see the revised abstract with (artificial) separation of each subsection:

(Background)

Withania chevalieri, endogenous from Cape Verde, is considered a medicinal plant, used in ethnomedicine with a large spectrum of applications, such as to treat skin fungal infections caused by dermatophytes.

(Objective)

The aim of this work was to chemical characterize W. chevalieri crude ethanolic extract (WcCEE), and to evaluate its bioactivities as antidermatophytic, antioxidant, anti-inflammatory and anticancer, as well as its cytotoxicity.

(Materials and Methods)

WcCEE was chemically characterized by HPLC-MS. Minimal inhibitory concentration, minimal fungicidal concentration, time-kill, and checkerboard assays were used to study the antidermatophytic activity of WcCEE. As an approach to the mechanism of action, cell wall components, b-1,3-glucan and chitin, and cell membrane ergosterol, were quantified; transmission electron microscopy (TEM) allowed the study of fungal ultrastructure.

(Results)

WcCEE contained phenolic acids, flavonoids and terpenes. It had a concentration-dependent fungicidal activity, not inducing relevant resistance, and is endowed with synergistic effects, especially with terbinafine. TEM showed severe damaged fungi; cell membrane and cell wall components levels had slight modifications. The extract had antioxidant, anti-inflammatory and anti-cancer activities, with low toxicity on non-tumoral cell lines.

(Conclusions)

The results demonstrated the potential of WcCEE as an antidermatophytic agent, with antioxidant, anti-inflammatory and anticancer activity, to be safely used in pharmaceutical and dermocosmetic applications.

Introduction: In this section, authors need to compare the purposes of the present study with similar studies existing in literature. These studies need an adequate presentation. Afterwards, authors need to clarify what their study brings in novelty. It is very important to state what exactly you bring in novelty in order to express your originality. The purpose of the study needs to be found in the last paragraph and be clearer. In fact, the whole section is unadequately organized: too many general informations in the first paragraph and too little specific ones in the second one. The specific part needs to be detailed. You need to offer informations on the state of the art of concepts that are included in your study. Please add further information and justifications and modify accordingly.

The authors acknowledge the reviewers comment.

Although the Introduction was not increased we introduced few details that we believe increase the readability of the introduction according to the reviewer recommendation, as follows:

-in the first paragraph are introduced general aspects about the use of natural extracts as therapeutics

- in the second part are introduced several aspects regarding other plants of the genus Whitania, the composition and the bioactivities,

- then it is introduced Whitania chevalieri as an endemic plant from Cape Verde and the fact that is used in ethnomedicine;

- and, finally, the novelty and objectives of our study was explained in the last paragraph of the Introduction.

Discussions: The first paragraph of this section is not adequately placed, it is more like an Introduction. Moreover, novelty and originality of your study is essential to be emphasized in this section once again. You need to emphasize this in terms of results, not purposes, as the Introduction should. You need to connect more the different concepts which are tested in this study.

The first paragraph was deleted.

The Discussion section is now initiated with the following paragraph:

(Line 322-324 clean MS; 357-359 marked MS) “In this work we characterized for the first time the composition of an ethanolic extract of W. chevalieri, an endemic plant of Cape Verde used in traditional medicine, and evaluated several bioactivities, such as, antidermatophytic, antioxidant, anti-inflammatory and anticancer.

Materials and Methods: You do not offer references for neither of the assays. Are all methods completely new? Because, if they are, you need to offer them a different approach.

The Materials and Methods were revised according to this reviewer’ comment. We tried to increase the readability of the methodologies to enable the reader, if needed, to replicate our methodologies. Some additional references were included whenever needed and removed if the methodology did not matched any other published before.

Conclusions: This section needs to be added, as Results and Discussion section is quite long. Please add perspectives of your study.

The section Conclusions was added as follows (Line 601-612 clean MS; 695-706 marked MS):

“5. Conclusions

Several Whitania spp. have been described as having bioactive properties. With this work it is reported for the first time the phytochemichal profile of a crude ethanolic extract of W. chevalieri (WcCEE), an endemic plant of Cape Verde archipelago. It can be concluded that WcCEE is an efficient inhibitor of the growth of dermatophytes, the fungi responsible for dermatophytoses, one of the most common skin infectious diseases. Moreover, to this antifungal/antidermatophytic activity, it adds that WcCEE synergizes with conventional antifungals used in human health to eradicate dermatophytoses, in particular with terbinafine. WcCEE also proved to be antioxidant, anti-inflammatory and anticancer, without having cytotoxic effects in non-tumoral cell lines. Overall, it can be concluded that there are scientific evidences supporting the efficiency and safeness of the W. chevalieri crude ethanolic extract prepared and used by Cape Verdeans in traditional medicine.”

References should follow the recommendations of the journal.

All the reference list was revised and corrected according to the Instructions to authors and the PLANTS template.

English needs serious revisions, if possible please review the whole manuscript, preferably by a native english speaker.

All these suggested changes should be performed in order to bring further improvements to the manuscript. 

All the reviewers pointed this aspect in the manuscript. This was one of our most important concerns when making the revision of the original manuscript. The entire manuscript was revised and modified to  improve language and grammar.

Round 2

Reviewer 1 Report

It is important to add in the conclusion that one of the perspectives of this work is the search for the active component (s) of the studied ethanolic extract.

Author Response

The authors acknowledge the reviewer's comment and added the following sentence to the "Conclusions" section:

(Line 649 – 652) “This work also opens the perspective of studying the bioactivities of individual compounds that we identified in the ethanolic crude extract of the whole W. chevalieri plant, envisaging novel therapeutic strategies.  

Reviewer 3 Report

Dear Authors,

The present study evaluates the antidermatophytic activity of a W. chevalieri crude ethanolic extract also and assesses its in vitro antioxidant, anti-inflammatory and cytotoxic effects. The authors performed most of the suggested changes after the first round of review. Following specific changes should still be performed in order to improve the quality of the manuscript:

Major changes

Introduction: I do not see the presentation of similar studies existing in literature. You need to compare your purposes with the ones of similar studies existing in literature and this will help you highlight the novelty and originality of your study. The specific part related to your study still needs to be detailed. Too many general aspects are presented and too few specific ones.

Discussions: Novelty and originality of your study needs to be emphasized in this section once again. You need to emphasize this in terms of results, not purposes, as the Introduction should.

Conclusions: Some perspectives should be added.

All these suggested changes should be performed in order to bring further improvements to the manuscript. 

English language needs moderate changes.

Author Response

Dear Reviewer

The authors acknowledge the reviewer's requests and comment and throughout the manuscript made modifications to fulfill these requests and comments. With this we believe that we improved the quality of the manuscript and for this we once again thank the reviewers for the careful revision.   Responses:

The present study evaluates the antidermatophytic activity of a W. chevalieri crude ethanolic extract also and assesses its in vitro antioxidant, anti-inflammatory and cytotoxic effects. The authors performed most of the suggested changes after the first round of review. Following specific changes should still be performed in order to improve the quality of the manuscript:

Major changes 

Introduction: I do not see the presentation of similar studies existing in literature. You need to compare your purposes with the ones of similar studies existing in literature and this will help you highlight the novelty and originality of your study. The specific part related to your study still needs to be detailed. Too many general aspects are presented and too few specific ones. 

Response:

The authors accepted this criticism of the reviewer regarding the Introduction section. To fulfill this request be the reviewer we added a reference to one of the last studies published in the literature as follows:

(Line 61-67) Although the medicinal use of some Whitania species, particularly W. somnifera,have been recognized for centuries, there is a constant update on the recognition of other species of this genus with interesting bioactivities. Recently, it was described th phytochemical composition of a root extract from Withania aristata (Aiton) Pauquy, endemic in the North African Sahara, and its strong antimicrobial activity against resistant bacteria and fungi, as well as its antioxidant and anti-inflammatory effects [15].

  1. Alzahrani, A.J. Promising Antioxidant and Antimicrobial Potencies of Chemically-Profiled Extract from Withania aristata (Aiton)Pauquy against Clinically-Pathogenic Microbial Strains. Molecules 2022, 27, 3614. https://doi.org/10.3390/molecules27113614

The novelty of our study our was developed in the Discussion section, since we based this not only in our purposes but also in the results obtained. Additional references describing specific studies similar to ours were added in the Discussion section.

(Line 411-417 – in what regards the anti-oxidant activity):

In this work, three methods were used to assess the antioxidative activity to contemplate the two criteria proposed by Laguerre et al. [55], and according to Phongpaichit et al. [56], the WcCEE showed a strong antioxidant activity. This result is according to what was reported in ethanolic extracts of root and leaves of Withania frutescens [39],  in ethanolic extracts of W. somnifera [7], hydro-ethanolic extract of W. aristata [15], and ethyl-acetate extracts of W. somnifera seeds [41], with higher antioxidant activity than aqueous or hydroethanolic extracts [7].

Two more references were added:

15.Alzahrani, A.J. Promising Antioxidant and Antimicrobial Potencies of Chemically-Profiled Extract from Withania aristata (Aiton)Pauquy against Clinically-Pathogenic Microbial Strains. Molecules 2022, 27, 3614. https://doi.org/10.3390/molecules27113614

  1. Mohsin, S.A.; Shaukat, S.; Nawaz, M.; Ur-Rehman, T.; Irshad, N.; Majid, M.; Hassan, SSu.; Bungau, S.; Fatima, H. Appraisal of selected ethnomedicinal plants as alternative therapies against onychomycosis: Evaluation of synergy and time-kill kinetics.Front Pharmacol. 2022, 13, 1067697. doi: 10.3389/fphar.2022.1067697

(Please see below other changes introduced in the Discussion section.)

Discussions: Novelty and originality of your study needs to be emphasized in this section once again. You need to emphasize this in terms of results, not purposes, as the Introductionshould.

Response:

This comment of the reviewer made us revise carefully what we believe is the novelties associated with our work and, based on that changes were introduced throughout the Discussion section. We strongly believe that the main novelties of this work are the following:

  1. It is the first time that Whitania chevalieri (a crude ethanolic extract) is studied in what regards its phytochemical profile and its bioactivities (anti-inflammatory, anti-oxidant, antifungal, specifically antidermatophytic). These innovative aspects were mentioned in the previous versions. This is the reason why he had to compare our results with those obtained by others in extracts from different species of the genus Whitania.

  1. The chemical characterization of chevalieri revealed ashagandhanolide e withanolid sulfoxide, only found before in Withania somnifera root. Before, baiantuoluoside J, was only found in Datura species.

(Line 342-348)

In general, the compounds present are similar to those found in Withania somnifera and other species of the genus Withania, however our study revealed the presence of ashagandhanolide and withanolid sulfoxide, previously found only in roots of Withania somnifera [29] and baiantuoluoside J, a norwithasteroid only found in Datura species [28]. Therefore, we consider that the results of the phytochemical characterization of WcCEE represents an important contribution to the expansion of knowledge about the species of the genus Withania.

  1. In this work the antifungal, specifically the antidermatophytic Effect was studied. This was done by many others before, with other Whitania species extracts. The novelty here is that we used several clinical isolates from diferente species of dermatophytes: 18 soudanense, 4 T. rubrum, 2 T.interdigitale and 2 M. canis.

(Line 363-366)

All the clinical isolates tested were susceptible to the antifungal activity of the WcCEE, with a MIC range of 1.56 - 25 mg/mL, even the strains resistant to griseofulvin. We used several clinical isolates of different species of dermatophytes, some of which isolated in Cape Verde.

  1. Our work describes for the first time an important aspect of the antimicrobial activity associated with novel antibiotherapeutics the study of resistance induction. As far as we could observe in other studies in Whitania plants extracts, this assay was never done before. The results were very interesting showing that the ethanolic extract we obtained (WcCEE) did not induce resistance.

(Line 386-391)

Our results also clearly demonstrate that WcCEE does not induces resistance after repeated exposure. In fact, this is an important issue when seeking for novel antimicrobial therapies, and we believe this is the first study demonstrating that an extract from plants of the genus Whitania do not induce resistance in fungi, specifically in dermatophytes. This might result from the chemical complexity of the extract acting on diverse targets, as described by others [46].

Conclusions: Some perspectives should be added.

Response:

The following sentence was added to the Conclusions section

(Line 649-652)

This work also opens the perspective of studying the bioactivities of individual compounds that we identified in the ethanolic crude extract of the whole W. chevalieri plant, envisaging novel therapeutic strategies.  

All these suggested changes should be performed in order to bring further improvements to the manuscript.